# Sex-Induced Changes in Microbial Eukaryotes and Prokaryotes in Gastrointestinal Tract of Simmental Cattle

**DOI:** 10.3390/biology13110932

**Published:** 2024-11-15

**Authors:** Diórman Rojas, Richard Estrada, Yolanda Romero, Deyanira Figueroa, Carlos Quilcate, Jorge J. Ganoza-Roncal, Jorge L. Maicelo, Pedro Coila, Wigoberto Alvarado, Ilse S. Cayo-Colca

**Affiliations:** 1Dirección de Desarrollo Tecnológico Agrario, Instituto Nacional de Innovación Agraria (INIA), Lima 15024, Peru; diormanr@gmail.com (D.R.); yolanda.bioinfo@gmail.com (Y.R.); deyanirafigueroa66@gmail.com (D.F.); ceqp2374@yahoo.com (C.Q.); jganoza@inia.gob.pe (J.J.G.-R.); 2Facultad de Ingeniería Zootecnista, Agronegocios y Biotecnología, Universidad Nacional Toribio Rodríguez de Mendoza de Amazonas (UNTRM), Cl. Higos Urco 342, Chachapoyas 01001, Peru; jmaicelo@untrm.edu.pe (J.L.M.); wigoberto.alvarado@untrm.edu.pe (W.A.); icayo.fizab@untrm.edu.pe (I.S.C.-C.); 3Facultad de Medicina Veterinaria y Zootecnia, Universidad Nacional del Altiplano de Puno, Puno 21001, Peru; pcoila@unap.edu.pe

**Keywords:** cattle gut microbiota, gender differences, archaea diversity, hematological parameters, livestock management

## Abstract

This study explored how the gut bacteria, archaea, and fungi differ between male and female Simmental cattle. By analyzing fecal samples, we found that certain types of microorganisms in the gut vary significantly depending on whether the cattle are male or female. For example, female cattle had a greater variety of certain archaea, which are microorganisms that help with digestion. Additionally, we discovered connections between specific gut microbes and blood health, with some microbes being more beneficial in males and others in females. These findings are important because they suggest that male and female cattle may need different feeding or health management strategies to optimize their growth and productivity. Understanding these differences can help farmers and veterinarians improve cattle health, which in turn could lead to better meat and milk production. This research emphasizes the importance of considering gender when studying the gut health of livestock.

## 1. Introduction

The ruminal microbiota is essential for the effective digestion of cellulosic compounds and other complex polysaccharides in cattle. Ruminants depend on a symbiotic relationship with the microorganisms present in the rumen, which produce enzymes necessary to break down these compounds into simpler molecules, facilitating their intestinal absorption. This specialized digestive system, consisting of the rumen, reticulum, omasum, and abomasum, has evolved to optimize the interaction of feed with the resident microflora [1,2,3]. The digestive capacity of livestock is closely linked to the activity of this microflora, which transforms food into essential nutrients for the animal. The composition of the microflora is crucial, as any alteration or the entry of pathogenic organisms can affect the animal’s metabolism and cause diseases [4]. Therefore, maintaining a proper rumen environment is vital for the health and productivity of cattle, especially in milk production.

Metabarcoding technology is essential for studying biodiversity, as it enables the identification of organisms through specific genes in environmental samples [5]. This technique, which utilizes advanced sequencing platforms such as those from Illumina, is crucial due to its high accuracy and deep sequencing capability [6]. As a result, it has revolutionized the characterization of microbial communities, including the gut mycobiota. The sequencing of marker genes has significantly expanded our understanding of the ecology and distribution of various organisms in different environments, from soils to the human gut, highlighting its relevance in ecological and health studies [7,8].

The sex of animals is a factor that can significantly influence the composition of the gut microbiota, with potential implications for the health and productive performance of livestock [9]. Hormonal variations between males and females affect the structure and diversity of the microbiota, modulating key processes such as digestion, metabolism, and immune response [10]. These differences can have important consequences for feed efficiency and disease susceptibility, highlighting the need to consider sex as a relevant variable in microbiota studies and in the design of nutritional management strategies to optimize gut health and performance in livestock [11]. Additionally, it has been shown that the gut microbiota plays a crucial role in modulating the levels of sex hormones, influencing the pathogenesis of various hormone-related diseases, such as ovarian cancer and polycystic ovary syndrome [12,13].

Consequently, this study aims to investigate the variations in the intestinal microbiota of cattle based on sex. The goal is to identify differences in the composition and function of the microbiota between males and females and assess how these differences may influence livestock productivity. A better understanding of these sex-related variations will enable the development of more effective management and feeding strategies, thereby optimizing intestinal functionality and productive performance in cattle. 

## 2. Materials and Methods

### 2.1. Sampling and Extraction of DNA

Previously, DNA samples were obtained in a prior study [14] Briefly, 21 fecal samples were collected from Simmental breed cattle at the Central Genetic Nucleus of the Donoso Agricultural Research Station (EEA Donoso) in Huaral, Lima, Peru. A preliminary study was carried out to identify the ideal number of replicates needed for the research. The cattle, divided into three age groups (58–63 months, 18–21 months, and 5 months), had a consistent sex ratio of 4 females to 3 males across all groups. All cattle were fed a diet with the same components, adjusted according to their age-specific needs (Appendix A). Fecal samples were obtained from the rectum, transported to the laboratory in liquid nitrogen, and stored at −80 °C for DNA extraction. Additionally, blood samples were taken from the jugular vein to detect hematological parameters (Appendix A). The cattle are part of a government-managed genetic nucleus, maintained under strict veterinary care to ensure health standards for semen and ovum donors, with no diseased animals present. The study adhered to Peruvian National Law No. 30407 on “Animal Protection and Welfare.” DNA was extracted from 21 fecal samples using the Stool DNA Isolation Kit (Norgen, Biotek Corporation, Sacramento, CA, USA) according to the manufacturer’s instructions. The concentration of the extracted DNA was quantified with a NanoDrop ND-1000 spectrophotometer, and the 260/280 absorbance ratio was measured to evaluate its quality. DNA integrity was assessed using 1% agarose gel electrophoresis.

### 2.2. PCR and Sequencing 

The DNA extracted from fecal samples was amplified using universal Archaeal primers Arch519F (5′CAGCCGCCGCGGTAA) and 519R (5′GTGCTCCCCCGCCAATTCCT), which are designed to target the V4–V5 variable regions of the 16S rRNA gene. For bacterial identification, DNA amplification was performed with primers 515F (GTGCCAGCMGCCGCGGTAA) and 806R (GGACTACHVGGGTWTCTAAT), which are specific to the V4 region of the 16S rRNA gene, generating a fragment of approximately 300 base pairs. For fungal identification, DNA was amplified using primers ITS3-2024F (GCATCGATGAAGAACGCAGC) and ITS4-2409R (TCCTCCGCTTATTGATATGC), which target the ITS2 region, producing an amplified fragment of about 380 base pairs. Duplicate PCR reactions were conducted and combined in equal volumes to ensure a sufficient amplicon quantity for Illumina Novaseq library preparations and to reduce PCR amplification bias. Sequencing libraries were generated using the Illumina TruSeq DNA PCR-Free Library Preparation Kit (Illumina, San Diego, CA, USA), following the manufacturer’s protocols. The quality of the prepared libraries was then assessed with a Qubit 2.0 Fluorometer (Thermo Scientific, Waltham, MA, USA). Finally, the validated libraries were sequenced on the Illumina NovaSeq 6000 platform (250 bp paired-end; Illumina Inc., San Diego, CA, USA) per the manufacturer’s guidelines.

### 2.3. Delimitation of Species

During the QIIME2 analysis, sequences were first subjected to trimming and quality control procedures. The paired-end reads, which were demultiplexed by Illumina, were then used to generate an amplicon sequence variant (ASV) table through the qiime2 [15] -dada2 [16] plugin software v2023.9. ASVs with fewer than 10 reads across all samples were excluded to minimize false positives, and sequences identified as plant or animal origin were also removed. Taxonomic classification was achieved using the SILVA v138.1 database for bacteria and archaea, while the UNITE ITS database was used for fungi. The high-quality sequences were aligned using MAFFT [17] within QIIME2. Rooted and unrooted phylogenetic trees for bacteria, archaea, and fungi were subsequently constructed using the FastTree algorithm available in the QIIME2 phylogenetic module.

### 2.4. Biostatical Analysis

The data analysis was conducted utilizing the Phyloseq [18] and Microeco [19] packages in R (v4.1.1) [20]. To determine the adequacy of sequencing depth, rarefaction curves were created for each sample. Alpha diversity metrics for intestinal bacteria, such as the observed species count (Observed), species richness estimate (ACE), Fisher’s index, and phylogenetic diversity (PD), were calculated, and a two-way ANOVA was applied to analyze the impact of age and sex. Beta diversity was explored using Jaccard and unweighted Unifrac distances, and the outcomes were displayed through principal coordinate analysis (PCoA). Group differences in bacterial communities were examined via two-way PERMANOVA [21], utilizing 9999 permutations for significance testing. Differential abundance of gut microbiota was assessed with edgeR [22]. Additionally, Spearman rank correlations were used to explore relationships between hematological parameters and alpha diversity indices, with the results visualized as heatmaps in R. Spearman’s rank correlation analyses, adjusted using false discovery rate (FDR) correction, were conducted to examine the associations between microbial genera and hematological parameters. The association between hematological variables and bacterial community composition was further investigated using Mantel tests with 999 permutations. Canonical correspondence analysis (CCA) was performed to assess the relationships between microbial community composition and environmental variables. Prediction of functional pathways was conducted using PICRUSt2 [23] to infer the metagenomic functional content from 16S rRNA gene sequences. The predicted functional profiles were then analyzed using STAMP v2.1.3 [24] (Statistical Analysis of Metagenomic Profiles). Differences in KEGG functional pathways between sexes were assessed using Welch’s two-sided *t*-test.

## 3. Results

### 3.1. Fecal Microbial Diversity in Sex 

Rarefaction curves for bacteria (Appendix A), archaea (Appendix A), and fungi (Appendix A) were constructed using the ASV minimum sample size. These curves demonstrated an adequate sampling depth, determined by the stabilization of the observed richness, without significant increases with increasing sequencing effort. Confidence intervals for each taxonomic group are presented in the corresponding Appendix A. Furthermore, Appendix A compares the richness between females and males, showing greater richness in females, which is supported by the confidence intervals shown in the Appendix A. No significant differences in alpha diversity were observed for bacteria and fungi (Appendix A). In contrast, alpha diversity analysis of archaea in the cattle gut microbiome indicated marked differences between females and males (Figure 1). The indices used included the observed species count (Observed, *p* = 0.029), which assesses total species richness; the ACE estimator (*p* = 0.028), which focuses on capturing rare species; Fisher’s index (*p* = 0.031), offering a robust perspective on diversity by considering relative abundance distributions; and phylogenetic diversity (PD, *p* = 0.01), which incorporates evolutionary relationships to provide an integrated view of community complexity. Across all indices, females consistently exhibited higher diversity than males. Additionally, a significant interaction between year and sex was detected for phylogenetic diversity (*p* = 0.046), suggesting a dynamic interplay between temporal and biological factors in shaping archaeal community structure.

For bacteria, the Jaccard distance-based PCoA (Figure 2A) identified a significant difference in community structure between females and males (*p* = 0.0483), suggesting that sex plays a role in shaping the bacterial composition, as indicated by PERMANOVA (Table 1). In contrast, the analysis of fungi using the Jaccard distance (Figure 2B) revealed a significant influence of year on community composition (*p* = 0.01). Similarly, when examining fungi with the unweighted Unifrac distance (Figure 2C), year was found to significantly impact the fungal community structure (*p* = 0.032). For archaea, the unweighted Unifrac distance-based PCoA (Figure 2D) demonstrated a significant difference in community structure between sexes (*p* = 0.0382), indicating a sex-specific effect on archaeal diversity.

The Venn diagrams (Appendix A) illustrate the distribution of amplicon sequence variants (ASVs) between female and male cattle for bacteria (Appendix A), fungi (Appendix A), and archaea (Appendix A). In bacteria, 3.3% of ASVs are unique to females, 2.2% are unique to males, and 94.5% are shared between the sexes. For fungi, 2.2% of ASVs are unique to females, 1.4% are unique to males, and 96.4% are shared. In archaea, 1% of ASVs are unique to females, 0% are unique to males, and 98.9% are shared. These results indicate a high degree of overlap in ASV composition between female and male cattle across all three microbial groups, with a small percentage of ASVs being sex-specific. Although the overlap is substantial, even small differences in ASV composition could still influence important physiological processes, such as metabolism or immune function.

The analysis of bacterial phyla in the gut microbiota of cattle revealed that the most dominant phyla were Firmicutes and Bacteroidota (Figure 3A), accounting for 55.61% and 31.15% of the total bacterial community in females, and 52.25% and 33.44% in males, respectively. Additionally, Verrucomicrobiota constituted 7.38% in females and 4.60% in males, while Spirochaetota made up 1.83% in females and 4.39% in males. Proteobacteria were also present at 1.50% in females and 3.07% in males. The remaining phyla each contributed less than 1% of the total bacterial composition. The analysis of fungal phyla (Figure 3B) in the gut microbiota of cattle revealed that the most dominant phylum was Ascomycota, accounting for 73.04% of the total fungal community in females and 83.89% in males. Mucoromycota constituted 21.42% of females and 13.30% of males. The unclassified fungal sequences (_k__Fungi) made up 5.06% in females and 2.54% in males. The remaining phyla each contributed less than 1% of the total fungal composition. The analysis of archaeal phyla (Figure 3C) in the gut microbiota of cattle revealed that the most dominant phyla were Euryarchaeota and Halobacterota, accounting for 29.47% and 63.40% of the total archaeal community in females, and 41.21% and 52.92% in males, respectively. Thermoplasmatota constituted 7.13% in females and 5.87% in males. 

The heatmaps (Figure 4) illustrate the relative abundance of genera within the gut microbiota of cattle, comparing female and male groups across bacteria, fungi, and archaea. In bacteria (Figure 4A), genera such as *UCG-005*, *UCG-010*, *Rikenellaceae RC9 gut group*, and *Solibacillus* are highly abundant in both sexes, while *Clostridium_sensu_stricto 1*, *Escherichia-Shigella*, and *Monoglobus* are among the least abundant. In fungi (Figure 4B), the genera *Candida/Metschnikowia*, *Kluyveromyces*, *Mucor*, and *Kurtzmaniella* are the most prevalent, whereas *Nakaseomyces*, *Wickerhamomyces*, and *Penicillium* exhibit lower abundance. In archaea (Figure 4C), *Methanocorpusculum* and *Methanobrevibacter* are the dominant genera, with *Methanosphaera* being less prevalent.

### 3.2. Effects of Sex on the Genera of the Intestinal Microbiota of Bovines

A Spearman correlation analysis was conducted to investigate the relationship between gut microbiota genera and hematological parameters in cattle, differentiating between females and males (Figure 5). In females, *UCG-010* exhibited a significant positive correlation with MCV, MCH, NEU%, SEG%, and TP. Succinivibrio correlated positively with MON, while *Prevotellaceae_UCG-004* was associated with WBC. *Prevotellaceae_UCG-001* demonstrated significant correlations, being positive with LYP and negative with *p-2534-18B5_gut_group*. The latter correlated positively with HCT and HGB, in contrast to Muribaculaceae, which correlated negatively with these same parameters. Mailhella was positively associated with HCT and HGB, while *M2PB4-65_termite_group* correlated negatively with NEU, SEG, NEU%, SEG%, and LYP. *Gastranaerophilales* exhibited a significant positive correlation with TP, and both *F082* and *dgA-11_gut_group* were positively associated with BAS and BAS%. *Christensenellaceae_R-7_group* correlated negatively with MCH, while *Bacteroides* was negatively associated with HCT and HGB. *Alloprevotella* demonstrated a significant positive correlation with PLT, and *Akkermansia* correlated negatively with EOS and EOS%. In males, WCHB1-41 correlated negatively with RBC, WBC, BAS, BAS%, and LYP, while *Victivallaceae* was negatively associated with BAS and BAS%. UCG-005 and UCG-002 exhibited significant positive correlations with RBC, WBC, BAS, BAS%, and LYP. *Treponema* demonstrated a significant positive correlation with MCH and MON, while *Solibacillus* correlated negatively with BAS and BAS%. *Rikenellaceae_RC9_gut_group* exhibited a positive correlation with MON%, whereas *RF39* correlated negatively with MON%. *Prevotellaceae_UCG-004* was positively associated with RBC, and *p-2534-18B5_gut_group* correlated negatively with PLT. *M2PB4-65_termite_group* demonstrated positive correlations with MCH and MON. *F082* correlated negatively with RBC, NEU, SEG, EOS, and TP, and positively with LYM and LYM%. *Escherichia-Shigella* was positively associated with PLT, while *dgA-11_gut_group* exhibited a significant negative correlation with EOS%. *Clostridia_vadinBB60_group* correlated negatively with EOS, MON%, and EOS%, while *Bacteroides* was negatively associated with NEU and SEG. Bacteroidales_RF16_group demonstrated positive correlations with MCV, MCH, MCHC, NEU, SEG, MON, EOS, MON%, and EOS%, and a negative correlation with LYM.

Regarding fungi (Figure 5B), in females, *Wickerhamiella* correlated positively with LYM% and PLT, while *Talaromyces* was positively associated with RBC. *Starmera* exhibited a significant negative correlation with MCH, and *Saturnispora* demonstrated a significant positive correlation with PLT. *Sarocladium* correlated negatively with TP, while *Rhizopus* exhibited a significant negative correlation with LYP. *Penicillium* was positively associated with PLT and negatively with LYP. *Naganishia* correlated positively with EOS and EOS%, while *Meyerozyma* demonstrated a negative correlation with PLT. *Magnusiomyces* was negatively associated with RBC, while *Kodamaea* exhibited negative correlations with MCV, MCHC, and TP. *Hannaella* correlated positively with MCH, while *Fusarium* demonstrated negative correlations with MCV, MCHC, NEU%, SEG%, and TP. *Cladosporium* was positively associated with PLT, while *Candida/Metschnikowiaceae* exhibited a significant negative correlation with MCV. *Aureobasidium* correlated negatively with NEU, SEG, and TP, and positively with LYM, LYM%, and PLT. *Aspergillus* demonstrated a significant positive correlation with PLT, while *Acremonium* exhibited positive correlations with RBC and negative correlations with TP. In males, *Zygoascus* exhibited significant negative correlations with MCV, MCHC, MON, and TP, and a positive correlation with LYM%. *Wickerhamiella* correlated negatively with NEU, SEG, and EOS, and positively with LYM and LYM%. *Sarocladium* demonstrated significant negative correlations with NEU, SEG, EOS, and EOS%, and positive correlations with LYM. *Saccharomyces* was positively associated with RBC, WBC, and LYM%, and negatively with TP. *Rhizopus* exhibited positive correlations with MCV, NEU, SEG, EOS, and TP, and negative correlations with LYM and LYM%. *Puccinia* was positively associated with RBC, WBC, and LYM%, and negatively with MCV, MCHC, MON, and TP. *Pichia* demonstrated a significant positive correlation with MCH and MON. *Pecoramyces* was positively associated with NEU% and SEG%. *Papiliotrema* exhibited significant negative correlations with MCV, MCHC, MON, EOS, MON%, and TP. *Nakaseomyces* correlated positively with MCV and MCHC. *Magnusiomyces* exhibited significant negative correlations with RBC and LYM, and positive correlations with NEU and SEG. *Kluyveromyces* demonstrated positive correlations with MCV, NEU, SEG, EOS, and TP, and negative correlations with LYM and LYM%. *Hanseniaspora* was negatively associated with NEU, SEG, EOS, and TP, and positively with LYM and LYM%. *Galactomyces* demonstrated positive correlations with MCV, MCHC, EOS, and TP, and negative correlations with LYM and LYM%. *Beauveria* exhibited significant negative correlations with MCV, MCHC, EOS, and TP. *Aspergillus* demonstrated significant positive correlations with NEU and SEG. Finally, *Anaeromyces* exhibited positive correlations with RBC, WBC, LYM%, and LYP, and negative correlations with MCV and TP. Regarding archaea (Figure 5C), in males, *Methanosphaera* exhibited significant negative correlations with MCV, MCHC, MON, and TP.

### 3.3. Differential Abundance Analysis of the Bovine Gut Microbiota

An edgeR analysis (Figure 6) was conducted to examine specific components of the gut microbiota in cattle, revealing significant differences by sex. In the female group, significant enrichment (*p* < 0.05) was identified in the following taxa: order (Coriobacteriales, Aeromonadales), Family (Rhizobiaceae, Succinivibrionaceae), genus (*Prevotellaceae_UCG-004*, *Erysipelotrichaceae UCG-008*, *Negativibacillus*, *UCG-010*, *Psychrobacillus*, *Pedobacter, Aeriscardovia, Falsochrobactrum, Turicibacter, Prevotella, Succinivibrio*), and species ([*Clostridium*] *methylpentosum*, Unclassified Species Oscillospiraceae, Unclassified Species *Enterorhabdus*, Unclassified Species *Ruminobacter*, Unclassified Species *Succinivibrio*) (Figure 6A). In the male group, significant enrichment (*p* < 0.05) was observed in the following taxa: order (Victivallales, Aeromonadales), class (Kiritimatiellae, Lentisphaeria), family (Victivallaceae, Atopobiaceae, Hafniaceae), genus (*WCHB1-41, vadinBE97, UCG-007*, Unclassified Genus Rhodobacteraceae, *Hafnia-Obesumbacterium*, *Faecalibacterium*, *CPla-4* termite group), and species (Unclassified Species Victivallaceae).

For fungi, significant enrichment (*p* < 0.05) was detected in the female group for the following taxa: order (Filobasidiales, Pleosporales), family (Filobasidiaceae, Sporormiaceae), genus (Y*amadazyma*, *Sepedonium*, *Naganishia*, *Preussia*), and species (*Candida ethanolica*, *Candida sinolaborantium*, *Sepedonium ampullosporum*, *Fusarium foetens*, Unclassified Species *Naganishia*, *Aspergillus mangaliensis*, *Pichia manshurica*) (Figure 6B). In the male group, significant enrichment was identified in the following taxa: order (Sordariales, Capnodiales), family (Capnodiaceae), genus (*Leptoxyphium*), and species: (Unclassified Penicillium, *Leptoxyphium glochidion*).

In methanogenic archaea, significant enrichment (*p* < 0.05) was noted in the female group for the following taxa: family (Methanosarcinaceae), genus (*Methanomicrococcus*, *Methanosarcinia*), and species (*Methanobrevibacter ruminantium*) (Figure 6C).

### 3.4. Microbial Community Dynamics

The Spearman correlation analysis presented (Figure 7) the relationships between alpha diversity indices and various hematological parameters in cattle (Appendix A). A Mann–Whitney test was conducted to assess the differences in hematological parameters between males and females. Significant differences based on sex were observed in leukocytes, neutrophils, segmented, lymphocytes, neutrophils (%), and lipids (Appendix A). In the fungal microbiota (Figure 7A), several significant correlations were identified. HGB had negative correlations with the Shannon and Pielou indices. MCV was negatively correlated with the Shannon and Pielou indices, while MCH was negatively correlated with the Shannon index. WBC had positive correlations with observed species richness, Chao1, and ACE. MON had negative correlations with the Shannon and Pielou indices. EOS and EOS% were negatively correlated with the Shannon and Simpson indices. TP had negative correlations with the Shannon, Simpson, and Pielou indices. In the archaeal microbiota (Figure 7B), HGB was negatively correlated with the Shannon index. WBC had positive correlations with observed species richness, Chao1, and ACE. MON% was positively correlated with observed species richness.

The Mantel and partial Mantel tests were conducted to assess the relationship between beta diversity and hematological parameters in cattle (Table 2). For bacteria, using the Jaccard distance, the Mantel test revealed significant correlations between beta diversity and both HCT (*p* = 0.005) and HGB (*p* = 0.022). The partial Mantel test confirmed these associations, with slightly adjusted *p*-values (HCT, *p* = 0.006; HGB, *p* = 0.021).

For fungi, using the weighted Unifrac distance, the Mantel test revealed a significant correlation between beta diversity and MCV (*p* = 0.022). The partial Mantel test further supported this relationship, with MCV also showing a significant correlation with beta diversity (*p* = 0.041). These results suggest that specific hematological parameters, such as HCT, HGB, and MCV, are associated with variations in the beta diversity of the bacterial and fungal communities in cattle, highlighting the potential impact of these hematological metrics on the gut microbiota composition.

Canonical correspondence analysis (CCA) (Appendix A) was conducted to explore the associations between microbial composition and hematological parameters, differentiated by sex. With respect to bacteria (Appendix A), females appeared to be more associated with parameters such as WBC, HGB, MCV, and MCH, whereas males showed a potential relationship with *Succinivibrionaceae_UCG-002* and *Clostridium sensu stricto 3*. Regarding fungi (Appendix A), females seemed to be linked to *Zaanenomyces*, which was associated with parameters like HGB and TP, while males displayed a possible association with *Cadophora* and variables such as PLT and NEU%. In the case of archaea (Appendix A), males demonstrated a notable association with methanogenic taxa, including *Methanobrevibacter* and *Methanosphaera*, while females exhibited correlations with MCH, MCV, and MON. 

The analysis performed with PICRUSt (Appendix A) revealed significant differences in several predicted metabolic functions between males and females. In males, higher proportions were identified in key metabolic pathways, such as purine nucleotide de novo biosynthesis (DENOVOPURINE2-PWY) and D-galacturonate degradation (GALACTUROCAT-PWY), suggesting increased microbial activity in these routes associated with nucleotide and carbohydrate metabolism. On the other hand, females showed higher proportions in pathways related to starch degradation (PWY-6731), geranylgeranyl diphosphate biosynthesis (PWY-5910), and photorespiration (PWY-181).

## 4. Discussion

The intestinal microbiota is essential for the productivity, health, and well-being of livestock, playing a key role in the regulation of digestive, metabolic, and even hematological processes [25,26]. It has been shown that the composition of the ruminal microbiota can be influenced by a variety of factors, such as diet, host type, breed, geographical location, and seasons [27,28]. In this study, significant differences in the alpha and beta diversity of intestinal archaea were found between females and males, suggesting a sex-specific effect on the structure of these microbial communities. These differences, particularly in phylogenetic diversity and in the composition of bacterial and archaeal communities, highlight the potential role of sex in the intestinal microbiota of cattle, which could influence the health and productive management of these animals.

In this study, the analysis of bacterial and fungal alpha diversity in the cattle gut microbiota revealed no significant differences related to age or gender, aligning with findings from other animal species. For example, recent studies in musk deer and Tibetan goats, as well as ruminants from the Tibetan Plateau, have similarly reported no significant differences in alpha diversity indices such as Chao1 and Shannon when comparing individuals of different genders and age groups [29,30,31]. The indices used in this study—Observed, ACE, Fisher, and PD—were specifically chosen to capture different aspects of microbial diversity: richness (Observed and ACE) and phylogenetic diversity (PD), with Fisher providing additional insight into community diversity structure. These indices were selected to provide a robust evaluation of microbial diversity across groups.

This suggests that age and gender may not substantially impact microbial diversity in bacteria and fungi, with factors such as diet and environment likely exerting a greater influence. The lack of significant findings for bacteria and fungi may reflect a more generalized response to environmental factors in these communities, contrasting with archaea, which could be more sensitive to intrinsic physiological factors, such as hormonal differences between sexes. The lack of significant differences in bacterial and fungal diversity by age and gender may be due to the fact that these communities are primarily influenced by external factors, such as diet and environment, which are often consistent among animals within the same management setting. Unlike bacteria and fungi, which are highly resilient and adaptable to host changes, methanogenic archaea respond more to intrinsic factors, such as hormone-related variations between genders, due to their specialized role in methanogenesis, which depends on hydrogen availability in the gut [32,33].

However, the current study did identify significant differences in archaea alpha diversity within the intestinal microbiome of cattle based on gender, consistent with observations in other species. This variation in archaeal diversity by sex could be attributed to physiological factors that uniquely influence archaeal communities, potentially through mechanisms associated with hormone regulation or energy metabolism. For instance, Ref. [34] reported notable differences in methanogenic archaea between young and adult reindeer in the Arctic tundra. Similarly, Ref. [35] found that the composition and diversity of microbial communities in the rumen during early life stages are significantly influenced by diet and environment. These findings underscore the importance of considering a multifactorial approach, where age, gender, and environmental factors act as key determinants in shaping the archaeal community composition in the gut microbiome.

Moreover, the study identified significant sex-based differences in the beta diversity of bacteria and archaea, indicating distinct microbial community compositions between males and females. This finding is consistent with previous research that has demonstrated gender-related differences in intestinal microbiota across various species. For instance, Ref. [36] observed distinct clustering of bacterial communities in bamboo rats based on gender, while [29] found similar patterns in cattle, reporting gender-specific differences in the beta diversity of bacteria and archaea. This supports the notion that physiological factors, such as hormonal influences, may shape microbial composition differently across sexes. Additionally, Ref. [37] noted the influence of age and gender on bacterial communities in Tibetan sheep, underscoring the complex role of biological factors in structuring microbiota.

Conversely, a significant influence of age on the beta diversity of intestinal fungi was observed, suggesting variability in fungal composition across different age groups. This result aligns with studies in other species, such as primates [38], where age-related fungal beta diversity was attributed to dietary changes and seasonal food availability, and human studies by [39], which identified age as a key factor in the mycobiota’s structure. These findings reinforce the importance of both intrinsic biological factors and external environmental influences, such as diet and seasonal changes, in shaping the composition of microbial communities.

Consistent with previous research, the dominant bacterial phyla identified in the cattle gut microbiota were Firmicutes and Bacteroidota [40,41,42,43,44]. The fungal community was predominantly composed of the phylum Ascomycota, which aligns with findings from other studies in cattle [45,46]. These results suggest a stable fungal community structure at the phylum level that is relatively unaffected by variations in environmental or dietary conditions [47,48].

In terms of archaeal composition, the identified phyla and genera were consistent with those reported in other ruminant studies. Similar to previous findings in cattle [35], the phylum Euryarchaeota and Halobacterota were predominant. This taxonomic pattern was also observed in other ruminants, such as moose, where [49] documented a similar predominance of these phyla in the rumen archaeal community. The consistency in the taxonomic composition of archaea and fungi across different ruminant species underscores the stability of these microbial groups in the intestinal microbiome, emphasizing the significant role of diet and environmental factors in shaping these communities.

The high overlap in ASVs between female and male cattle across bacteria, fungi, and archaea suggests a shared core microbiome that likely supports essential microbial functions common to both sexes [50]. Despite this similarity, the presence of a small subset of sex-specific ASVs, although limited in percentage, may contribute to distinct biological roles that align with sex-specific metabolic or immune processes [51]. Such differences, while subtle, could influence physiological traits unique to each sex, underscoring the potential relevance of even minor variations in microbial composition [52]. Further research may elucidate how these unique ASVs impact broader physiological or metabolic pathways. The significant differences in hematological parameters between sexes suggest variations in immune activity and metabolic profile, which could be related to specific physiological demands or differences in hormonal regulation between females and males [53,54]. These findings highlight the potential impact of sex on health and response to external conditions, justifying additional studies to better understand these influences.

The correlation observed in this study between *Treponema* and the hematological parameters MCH and MON in males suggests that an intestinal microbiota rich in *Treponema* could be associated with a more favorable hematological profile, which would have significant implications for the general health and productive performance of male bovines. The genotypic and phenotypic diversity of *Treponema* in the bovine gastrointestinal tract indicates that different *Treponema* phylotypes could play specific roles, possibly as beneficial commensals [55]. These roles could include modulation of the immune response and optimization of nutrient metabolism, which in turn would positively impact the observed hematological parameters, such as MCH and MON. Recent research has highlighted that the abundance of *Treponema* in the rumen of bovines is associated with greater efficiency in nutrient metabolization [56]. Additionally, although *Treponema* has been identified as a prevalent pathogen in certain bovine lesions related to lameness, it has been suggested that its presence in these cases could be more opportunistic than causal [57]. This reinforces the idea that, in the intestinal context, *Treponema* could be playing a more beneficial or neutral role, contributing to the stability and functionality of the microbial ecosystem, which in turn is reflected in favorable hematological parameters [56]. Finally, studies in traditional human populations have demonstrated the significant presence of *Treponema* in the intestinal microbiota, suggesting a possible role in the modulation of intestinal health and, potentially, in the regulation of hematological functions [58].

The significant negative correlation between *Akkermansia* and EOS and EOS% parameters in females observed in this study could be associated with the role of *Akkermansia* as a mucinolytic bacterium, which degrades mucin in the intestinal barrier, potentially affecting barrier integrity and influencing hematological parameters [59]. Recent investigations have identified the important role of eosinophils in the regulation of mucosal microbiota, suggesting that alterations in bacterial diversity within the intestinal mucosa may have considerable implications for intestinal homeostasis [60]. This influence on resident microbiota may account for the observed relationship in this study, indicating that the presence of *Akkermansia* in the intestinal mucosa of females could be associated with reduced eosinophil activation, reflecting changes in local hematological parameters. Additionally, *Akkermansia* has been typically detected in the feces of cattle and is abundant in those fed a high-forage diet, underscoring its role in the degradation of the mucosal layer [9,61].

Studies indicate that *Saccharomyces boulardii* modulates the intestinal microbiota, reducing systemic and hepatic inflammation in models of obesity and type 2 diabetes, and enhancing the intestinal barrier function in cases of induced colitis [62,63,64,65]. In species like horses and cattle, *Saccharomyces cerevisiae* also impacts the microbiota, modulating bacteria associated with fiber fermentation and lactate stabilization [66,67]. In this study, a positive correlation was identified between *Saccharomyces* and RBC, WBC, and LYM% counts, with a negative correlation with protein levels, suggesting that this genus may influence the intestinal microbiota and impact hematological parameters essential to cattle health [68].

Aspergillus, a common genus in the gastrointestinal tract of various animals, is linked to fiber digestion and microbial balance preservation, especially in the absence of pathogenic factors. Here, a significant positive correlation emerged between Aspergillus and platelet counts, suggesting an indirect role in regulating hematological parameters [69,70]. *Methanosphaera*, recognized for its role in methanogenesis through methanol reduction, contributes to methane production in ruminants, primarily in the solid fraction of the rumen [71,72]. In this analysis, a significant negative correlation was observed between *Methanosphaera* and values of MCV, MCH, MON, and protein levels in males, potentially indicating an adverse impact on host health and metabolism. This highlights the importance of investigating complex interactions between methanogens and physiological parameters [73,74].

The identification of *Prevotellaceae_UCG-004* in bulls underscores its role in ruminal fermentation and volatile fatty acid (VFA) production, particularly acetate and butyrate. Prior studies associate *Prevotellaceae_UCG-004* with improved growth performance in high-body-weight goat kids, attributed to its ability to degrade complex polysaccharides and support ruminal epithelium development [75]. Additionally, its association with ascorbate and aldarate metabolism suggests that a high dietary vitamin C content could enhance its presence in the ruminal microbiota, optimizing energy metabolism and productive performance in cattle [76]. The co-occurrence of genera such as *Methanobrevibacter*, *Prevotellaceae_UCG-004*, and Treponema suggests an interdependent microbial network that optimizes the rumen’s anaerobic environment, promoting VFA production essential for cattle growth and performance. These mutualistic interactions, especially between *Methanobrevibacter* and fiber-degrading bacteria, highlight opportunities for dietary interventions to improve rumen health and potentially reduce methane emissions [71,75,76].

The presence of *Negativibacillus* in the bull microbiota could indicate a robust microbiome that supports overall well-being. Although associated with dysbiosis in other contexts, in healthy animals, it may contribute to intestinal homeostasis and productive performance, with additional positive effects on reproductive health [77,78].

The detection of *Naganishia* in bulls suggests a role in rumen health and modulation of the ruminal microbial environment. Prior studies have identified *Naganishia* in young ruminants, where it may support fiber digestion and microbial balance, contributing to rumen ecosystem stability [79,80].

The detection of the genus *Preussia* in bulls is notable for its antibacterial and antioxidant properties, which could contribute to disease resistance [81,82]. In previous studies, *Preussia* has also been found in the intestinal microbiota of yaks, suggesting a role in adapting to high-altitude environments and improving intestinal health [83]. Its presence in bulls reinforces its possible contribution to the well-being and productive performance of cattle.

*Methanobrevibacter ruminantium*, identified in females, is associated with lower methane emissions compared to *Methanobrevibacter gottschalkii*, potentially indicating an adaptation of the ruminal microbiota towards optimized hydrogen and formate utilization in methane metabolism [84,85,86].

In this study, a significant negative correlation was observed between total protein content and fungal alpha diversity, measured by the Shannon, Simpson, and Pielou indices. These findings suggest that higher protein intake could be associated with a reduction in intestinal fungal diversity, which may influence the stability and functionality of the fungal microbiome. This effect could be related to the impact of proteins on resource availability and microbial competition in the intestine, thus affecting the composition and diversity of the fungal community [87].

The observed correlation between leukocytes and fungal and archaeal alpha diversity in our study suggests a possible interaction between the immune response and microbial composition in the bovine intestine. Although the context is different, studies in humans have indicated that alterations in leukocyte levels can influence intestinal microbial diversity, as observed in patients with colorectal cancer [88].

Mantel and partial Mantel tests revealed significant correlations between bacterial beta diversity and levels of HCT and HGB in healthy cattle. These findings suggest an intrinsic relationship between the composition of the intestinal microbiota and the hematological status of the animals [89]. The positive correlation between HCT, HGB, and intestinal microbial diversity could indicate that a favorable hematological profile is associated with a diverse and balanced microbiota, which is crucial for maintaining systemic homeostasis and digestive health in ruminants [90]. The importance of microbial diversity has been emphasized in previous studies, which have identified differences in alpha and beta diversity in individuals with various conditions [91]. This approach is essential in studies involving healthy cattle, as it facilitates the evaluation of how bacterial diversity may relate to physiological or management parameters, thereby helping to identify practices that optimize intestinal health and animal productivity [92].

Furthermore, a significant relationship was identified between mean corpuscular volume (MCV) and fungal beta diversity, suggesting that variations in red blood cell volume may be related to changes in gut mycobiota composition. Clinical indicators such as MCV have been associated with alterations in the diversity and prevalence of certain fungal species [93]. This connection underscores the importance of investigating how variations in clinical hematological parameters could influence the diversity and functionality of the intestinal mycobiota, with implications for maintaining intestinal balance and overall systemic health [94].

In the CCA analysis, the differences observed between healthy males and females in the parameters EOS, EOS%, and BAS% suggest that the intestinal microbiota could play a key role in the regulation of inflammation and immune response. In males, higher levels of eosinophils and basophils could be linked to immune modulation, possibly through the interaction between the microbiota and the immune system, which regulates inflammatory mediators and maintains immune homeostasis [95,96]. In contrast, females showed a stronger relationship with fermentative bacteria such as *Succinivibrionaceae_UCG-002* and *Clostridium sensu stricto 3*, known for their role in the production of short-chain fatty acids (SCFAs), suggesting an adaptation of their microbiota towards energy metabolism and fermentation [97,98]. These differences indicate a possible metabolic and immunological specialization influenced by sex and the intestinal microbiota in cattle.

The increased activity of the GALACTUROCAT-PWY pathway in male cattle suggests a microbiological adaptation that enhances the breakdown of complex carbohydrates such as pectin, optimizing energy metabolism through the production of short-chain fatty acids (SCFAs), which are essential for intestinal and energy homeostasis [99,100]. This intensified activity in males may reflect a physiological adjustment of the microbiota to meet sex-specific energy demands, promoting gut health and nutrient use efficiency [101]. The increased activity of the PWY-5265 pathway (peptidoglycan biosynthesis) in female cattle suggests an adaptation that enhances bacterial cell wall stability, supporting immune regulation and mucosal integrity, and promoting a more resilient microbiota [102,103]. The presence of *Faecalibacterium* in females aligns with this function, given its role in butyrate production and immune modulation, which contribute to intestinal health. Previous studies associate *Faecalibacterium* prevalence with improved weight gain and reduced diarrhea in early life stages, underscoring its probiotic potential [104,105,106]. The detection of *Faecalibacterium* here highlights its positive impact on metabolic balance and welfare in female cattle.

Although this study did not directly measure hormone levels, the observed gender-based differences in microbial diversity and composition suggest that hormonal factors may play a significant role in shaping the gut microbiome in cattle [11]. Sex hormones such as estrogen and testosterone are known to influence various physiological processes, including the modulation of microbial communities [107]. Previous research has demonstrated that these hormones can affect the metabolism and immune interactions of gut microbiota, potentially leading to differences in microbial structure between males and females [108]. The distinct clustering of bacterial and archaeal communities observed in this study could therefore be partially attributed to the differential effects of sex hormones, which might influence the composition and functional dynamics of the intestinal microbiome [109]. Future studies integrating hormonal data could provide deeper insights into the mechanisms driving these gender-specific microbial patterns.

## 5. Conclusions

This study revealed significant gender-based differences in the gut microbiota of bovines, with distinct patterns in bacterial, archaeal, and fungal communities between males and females. The observed negative correlation between *Methanosphaera* and key hematological parameters in males suggests its influence on metabolic processes, which could impact health and performance. Additionally, the association of Treponema with favorable hematological profiles in males and the correlation of *Akkermansia* with hematological variations in females emphasize the relevance of gender in shaping the microbiota. These insights underscore the potential of tailoring microbiota management strategies based on gender to optimize cattle health and productivity, representing a promising approach for enhancing performance and efficiency in the cattle farming industry.

## Figures and Tables

**Figure 1 biology-13-00932-f001:**
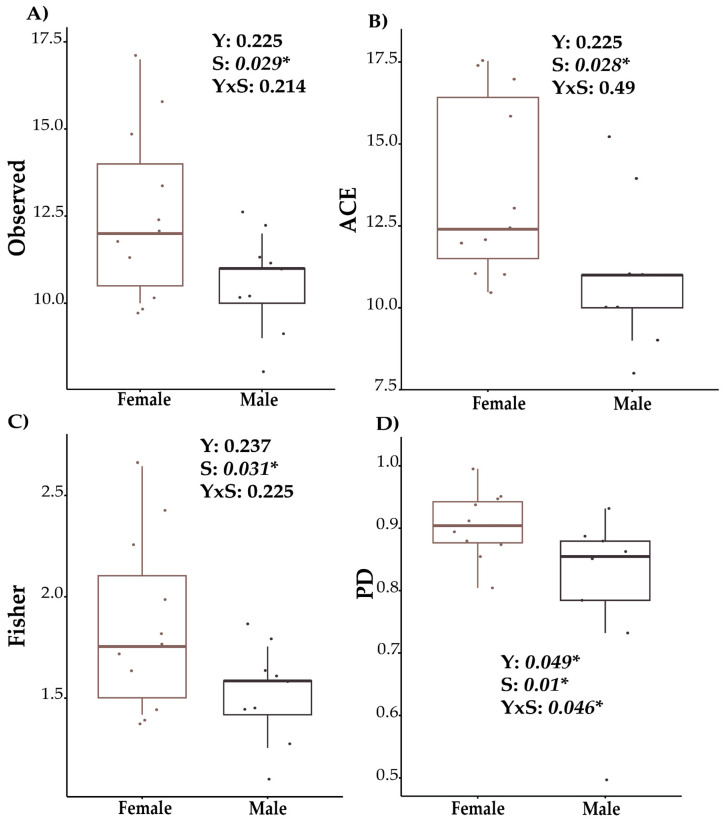
Comparison of alpha diversity of archaea in the cattle gut microbiome between females and males. Y = year, S = sex. YxS = Year x Sex (**A**) Observed. (**B**) ACE. (**C**) Fisher. (**D**) PD. * *p* < 0.05.

**Figure 2 biology-13-00932-f002:**
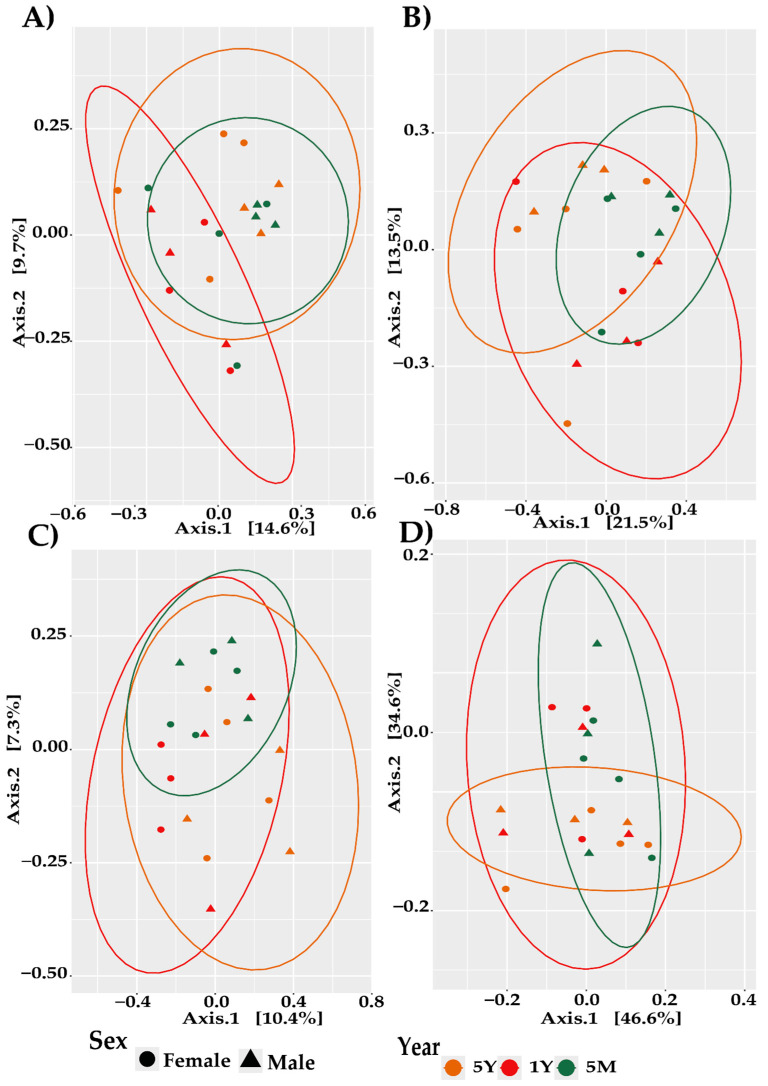
PCoA plots of beta diversity in cattle gut microbiota based on sex. (**A**) Bacteria (Jaccard distance). (**B**) Fungi (Jaccard distance). (**C**) Fungi (unweighted Unifrac). (**D**) Archaea (unweighted Unifrac).

**Figure 3 biology-13-00932-f003:**
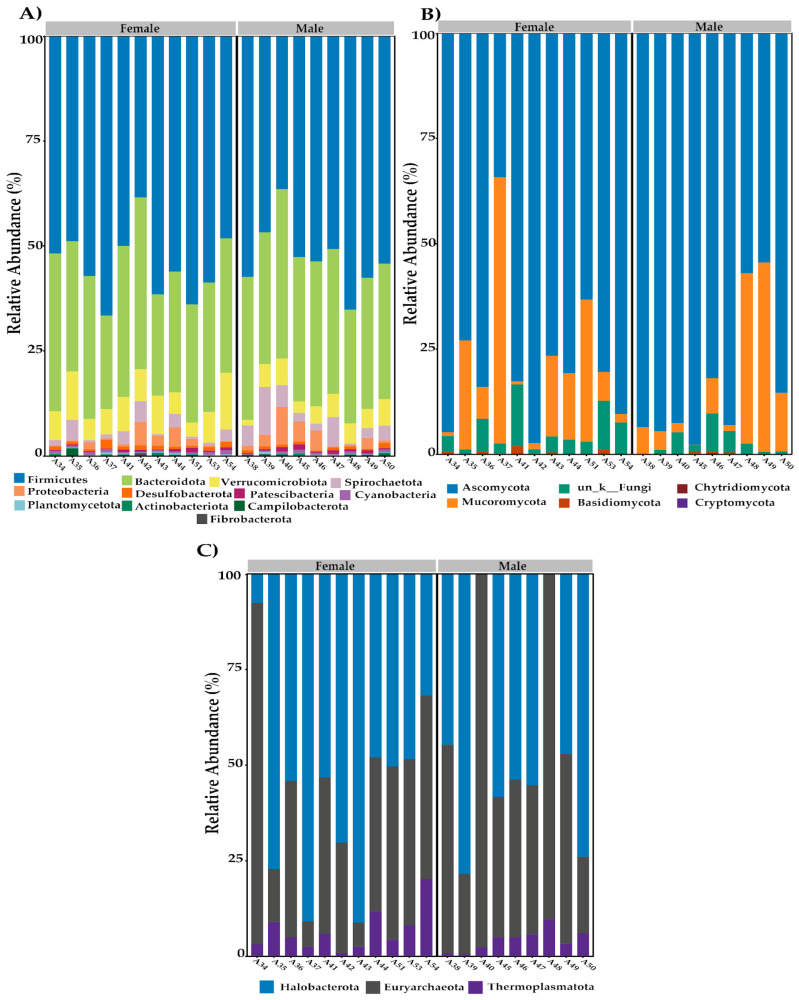
Relative abundance of microbial phyla in the gut microbiota of cattle by sex. (**A**) Bacteria. (**B**) Fungi. (**C**) Archaea.

**Figure 4 biology-13-00932-f004:**
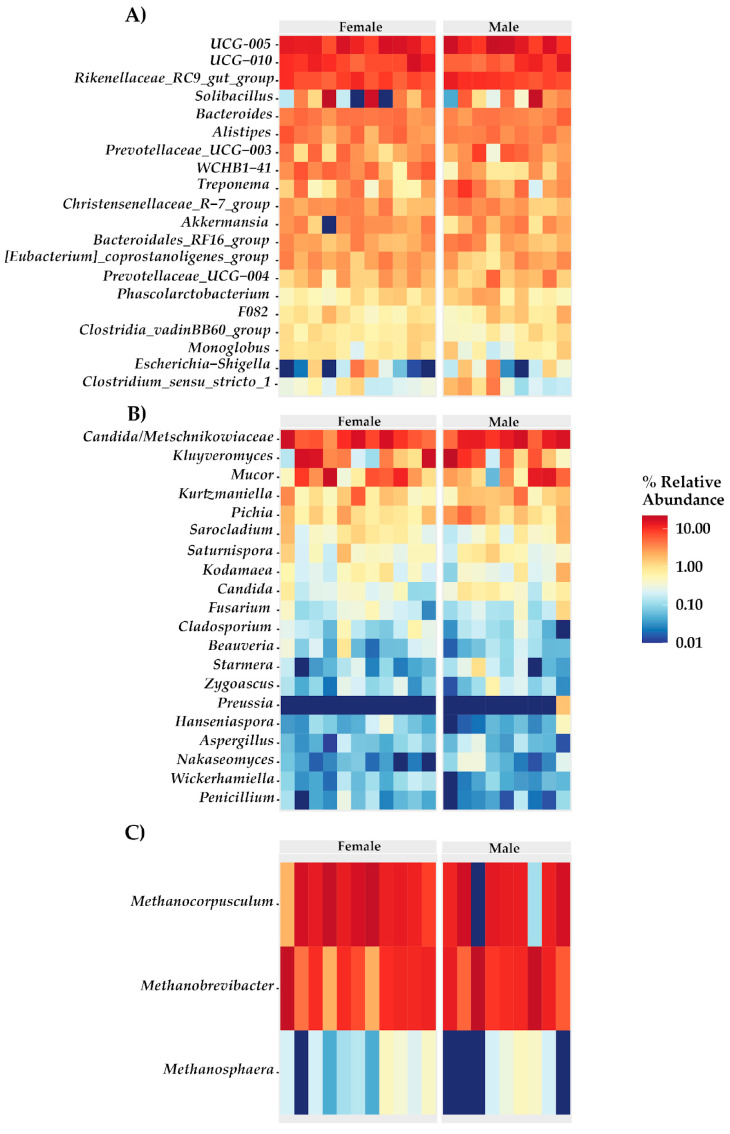
Heatmaps of the relative abundance of microbial genera in the gut microbiota of cattle by sex. (**A**) Bacteria. (**B**) Fungi. (**C**) Archaea.

**Figure 5 biology-13-00932-f005:**
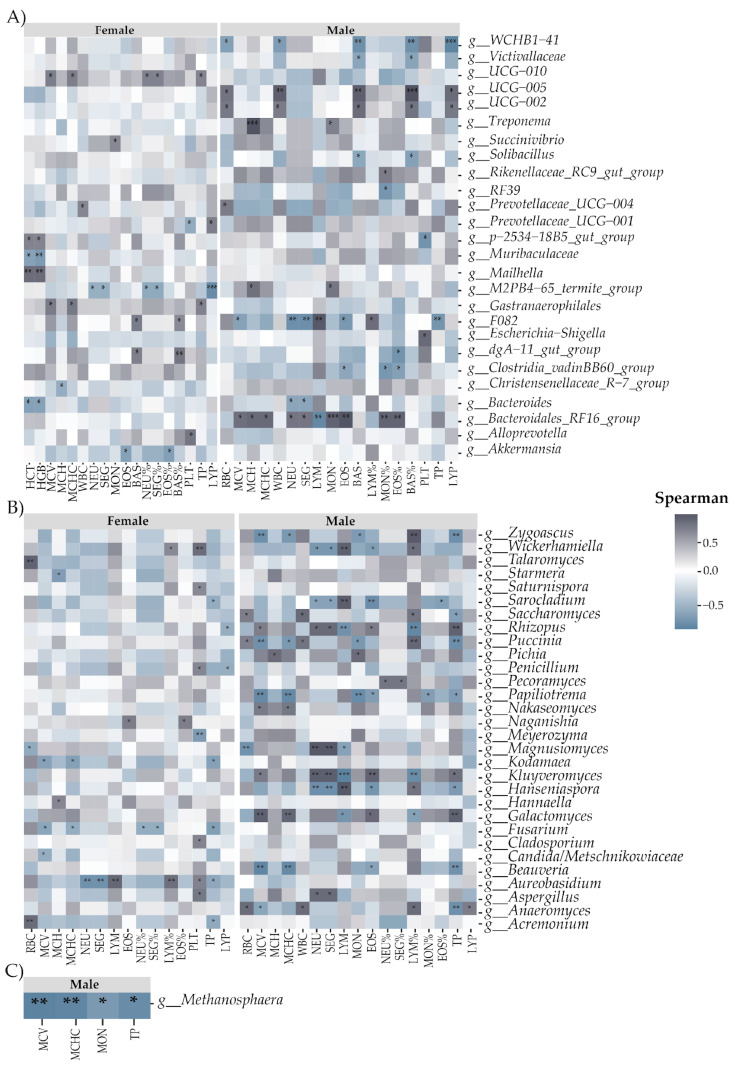
Spearman correlation between gut microbiota and hematological parameters in female and male cattle. (**A**) Bacteria. (**B**) Fungi. (**C**) Archaea. * *p* < 0.05, ** *p* < 0.01, *** *p* < 0.001.

**Figure 6 biology-13-00932-f006:**
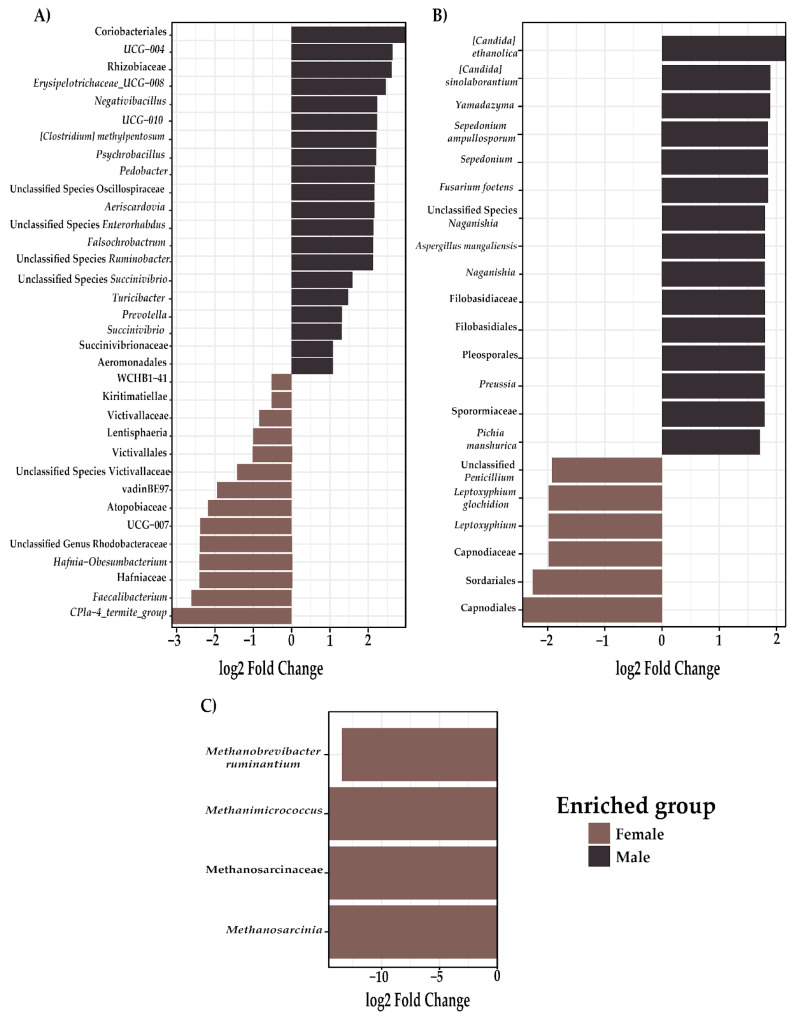
Differential abundance analysis of the bovine gut microbiota by sex. (**A**) Bacterial taxa. (**B**) Fungal taxa. (**C**) Archaeal taxa. Log2 fold changes indicate significant enrichment (*p* < 0.05) in male (brown) or female (black) groups.

**Figure 7 biology-13-00932-f007:**
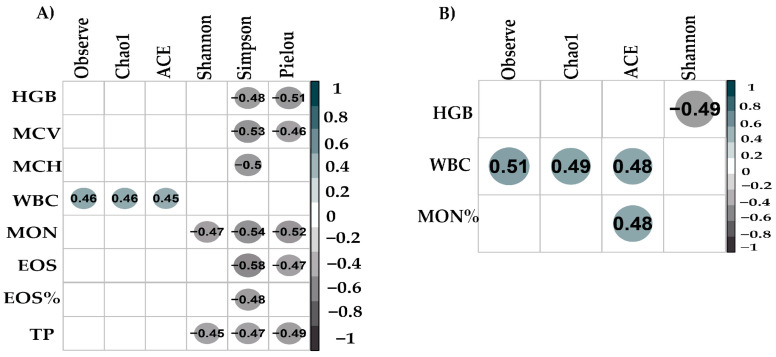
Spearman correlations between alpha diversity indices and hematological parameters in cattle. (**A**) Fungi. (**B**) Archaea.

**Table 1 biology-13-00932-t001:** PERMANOVA of unweighted Unifrac and Jaccard methods * *p* < 0.05, ** *p* < 0.01.

	Items	Df	Sum of Sqs	R^2^	F	Pr (>F)
Bacteria	Jaccard					
	Year	2	0.5312	0.11634	1.1475	0.0632
	Sex	1	0.2852	0.06247	1.2323	0.0483 *
	Year/sex	2	0.509	0.11149	1.0997	0.125
	Residual	14	3.2403	1		
	Total	19	4.5657			
Fungi	Jaccard					
	Year	2	0.7816	0.11981	1.1786	0.0099 **
	Sex	1	0.3715	0.05694	1.1203	0.069
	Year/sex	2	0.7287	0.1117	1.0989	0.055
	Residual	14	4.6421	0.71155		
	Total	19	6.5239	1		
	Unweighted Unifrac					
	Year	2	0.16777	0.18907	2.0254	0.032 *
	Sex	1	0.04423	0.04985	1.0679	0.374
	Year/sex	2	0.0955	0.10763	1.1529	0.323
	Residual	14	0.57982	0.65345		
	Total	19	0.88731	1		
Archaea	Unweighted Unifrac					
	Year	2	0.05007	0.12172	1.3689	0.201
	Sex	1	0.04789	0.11644	2.6188	0.0382 *
	Year/sex	2	0.05733	0.13937	1.5673	0.1322
	Residual	14	0.25603	0.62247		
	Total	19	0.41132	1		

**Table 2 biology-13-00932-t002:** Beta diversity and hematological parameters: Mantel and partial Mantel test results.

Bacteria	Jaccard			
	Mantel Test		Partial Mantel Test	
Variables	r	*p*	r	*p*
HCT	0.376950553	0.005	0.36636155	0.006
HGB	0.342372916	0.022	0.326735683	0.021
**Fungi**	**Weighted Unifrac**			
	**Mantel Test**		**Partial Mantel Test**	
Variables	r	*p*	r	*p*
MCV	0.3124	0.022	0.170249895	0.041

## Data Availability

The raw data supporting the conclusions of this article will be made available by the authors on request.

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
