# Peer review of "Sex-Induced Changes in Microbial Eukaryotes and Prokaryotes in Gastrointestinal Tract of Simmental Cattle"

_biology, 2024, doi:10.3390/biology13110932_

Round 1
Reviewer 1 Report
Comments and Suggestions for Authors
This study revealed gender-specific differences in the gut microbiota of Simmental cattle, with females showing greater archaeal diversity and distinct microbial compositions. The findings suggest that gender-specific feeding and health strategies could optimize cattle growth and productivity.
(Lines 146–148) The text states that the rarefaction curves demonstrated sufficient sampling depth, but it lacks quantitative information (like sequencing depth or a threshold for sufficiency). There is no explicit mention of how this sufficiency was determined, nor are confidence intervals or error bars described.
(Lines 149–155) The results focus only on significant findings for archaea, but there is no discussion about why alpha diversity might differ by sex. The lack of significant differences for bacteria and fungi is briefly mentioned but not explored further. This section also lacks details about why specific diversity metrics (Observed, ACE, Fisher, PD) were chosen.
(Lines 161–169)The significant differences in beta diversity are presented, but no biological context is provided to interpret these findings. Additionally, the narrative jumps between taxa (bacteria, fungi, archaea) without connecting these results to potential sex-based differences in the gut microbiome composition.
(Lines 178–184)The percentages of unique and shared Amplicon Sequence Variants (ASVs) between sexes are given, but the biological implications of these percentages are not discussed. The overlap of 94.5% in bacteria, 96.4% in fungi, and 98.9% in archaea suggests minimal sex-specific differences, yet these findings aren't critically analyzed.
(Lines 187–188)The p-values and F statistics presented are borderline significant (especially for sex in bacteria and archaea), but the power of the study (sample size) is not addressed. It’s possible that the results could be underpowered, which might explain the lack of significance in some cases.
(Lines 213–221) The description of genera is purely descriptive without an interpretation of what the differences might mean in terms of functionality or cattle physiology. The presentation is also slightly redundant with the earlier section on phyla.
(Lines 223–289) The text lacks a critical assessment of whether these correlations are robust or just statistically significant by chance.
Evaluate the study’s power to detect significant differences, especially where borderline significance is observed.
Conduct multivariate analyses (e.g., Canonical Correspondence Analysis) to better understand how environmental or physiological factors interact with microbial diversity and composition.
Include functional predictions (e.g., through PICRUSt2) to see how the observed microbial shifts might translate to functional changes in metabolism or immune function.
Author Response
This study revealed gender-specific differences in the gut microbiota of Simmental cattle, with females showing greater archaeal diversity and distinct microbial compositions. The findings suggest that gender-specific feeding and health strategies could optimize cattle growth and productivity.
> Thank you for your comment. We agree that gender-specific microbiota differences are relevant for optimizing feeding and health strategies in cattle.
(Lines 146–148) The text states that the rarefaction curves demonstrated sufficient sampling depth, but it lacks quantitative information (like sequencing depth or a threshold for sufficiency). There is no explicit mention of how this sufficiency was determined, nor are confidence intervals or error bars described.
> Thank you for your observation. The explanation regarding the determination of sufficient sampling depth has been added to the manuscript.
In results, we added a paragraph from the L165 to 173 “Rarefaction curves for bacteria (Fig. S1A), archaea (Fig. S1B), and fungi (Fig. S1C) were constructed using the ASV minimum sample size. Confidence intervals for each taxonomic group are presented in the corresponding supplementary tables (Table S2 for bacteria, Table S3 for fungi, and Table S4 for archaea). Furthermore, Supplementary Figure 2 compares the richness between females and males, showing greater richness in females, which is supported by the confidence intervals shown in the supplementary tables.”
-Tessler, M., Neumann, J. S., Afshinnekoo, E., Pineda, M., Hersch, R., Velho, L. F. M., ... & Brugler, M. R. (2017). Large-scale differences in microbial biodiversity discovery between 16S amplicon and shotgun sequencing. Scientific reports, 7(1), 6589.
-Beule, L., & Karlovsky, P. (2020). Improved normalization of species count data in ecology by scaling with ranked subsampling (SRS): application to microbial communities. PeerJ, 8, e9593.
-Lin, H., & Peddada, S. D. (2020). Analysis of microbial compositions: a review of normalization and differential abundance analysis. NPJ biofilms and microbiomes, 6(1), 60.
(Lines 149–155) The results focus only on significant findings for archaea, but there is no discussion about why alpha diversity might differ by sex. The lack of significant differences for bacteria and fungi is briefly mentioned but not explored further. This section also lacks details about why specific diversity metrics (Observed, ACE, Fisher, PD) were chosen.
> Thank you for your suggestions. We have added explanations in the results and discussion regarding the selection of alpha diversity indices (Observed, ACE, Fisher, PD).
In results, we added a paragraph from the L176 to 187: "No significant differences in alpha diversity were observed for bacteria and fungi (Table S3). In contrast, alpha diversity analysis of archaea in the cattle gut microbiome indicated marked differences between females and males. The indices used included the observed species count (Observed, p = 0.029), which assesses total species richness; the ACE estimator (p = 0.028), which focuses on capturing rare species; Fisher's index (p = 0.031), offering a robust perspective on diversity by considering relative abundance distributions; and phylogenetic diversity (PD, p = 0.01), which incorporates evolutionary relationships to provide an integrated view of community complexity. Across all indices, females consistently exhibited higher diversity than males. Additionally, a significant interaction between year and sex was detected for phylogenetic diversity (p = 0.046), suggesting a dynamic interplay between temporal and biological factors in shaping archaeal community structure."
In the discussion, we added in the L439-463: "In this study, the analysis of bacterial and fungal alpha diversity in the cattle gut microbiota revealed no significant differences related to age or gender, aligning with findings from other animal species. For example, recent studies in musk deer and Tibetan goats, as well as ruminants from the Tibetan Plateau, have similarly reported no significant differences in alpha diversity indices such as Chao1 and Shannon when comparing individuals of different genders and age groups [26–28]. The indices used in this study—Observed, ACE, Fisher, and PD—were specifically chosen to capture different aspects of microbial diversity: richness (Observed and ACE) and phylogenetic diversity (PD), with Fisher providing additional insight into community diversity structure. These indices were selected to provide a robust evaluation of microbial diversity across groups. This suggests that age and gender may not substantially impact microbial diversity in bacteria and fungi, with factors such as diet and environment likely exerting a greater influence. The lack of significant findings for bacteria and fungi may reflect a more generalized response to environmental factors in these communities, contrasting with archaea, which could be more sensitive to intrinsic physiological factors, such as hormonal differences between sexes. The lack of significant differences in bacterial and fungal diversity by age and gender may be due to the fact that these communities are primarily influenced by external factors, such as diet and environment, which are often consistent among animals within the same management setting. Unlike bacteria and fungi, which are highly resilient and adaptable to host changes, methanogenic archaea respond more to intrinsic factors, such as hormone-related variations between genders, due to their specialized role in methanogenesis, which depends on hydrogen availability in the gut (Donaldson et al., 2016; Evans et al., 2020)."
-Sommer, F., & Bäckhed, F. (2013). The gut microbiota—masters of host development and physiology. Nature reviews microbiology, 11(4), 227-238.
-Ungerfeld, E. M. (2020). Metabolic hydrogen flows in rumen fermentation: principles and possibilities of interventions. Frontiers in Microbiology, 11, 589.
(Lines 161–169)The significant differences in beta diversity are presented, but no biological context is provided to interpret these findings. Additionally, the narrative jumps between taxa (bacteria, fungi, archaea) without connecting these results to potential sex-based differences in the gut microbiome composition.
> Thanks for your comments. We have added biological context in the discussion section to interpret the differences in beta diversity observed. In the L487-506 “Moreover, the study identified significant sex-based differences in the beta diversity of bacteria and archaea, indicating distinct microbial community compositions between males and females. This finding is consistent with previous research that has demonstrated gender-related differences in intestinal microbiota across various species. For instance, [31] observed distinct clustering of bacterial communities in bamboo rats based on gender, while [26] found similar patterns in cattle, reporting gender-specific differences in the beta diversity of bacteria and archaea. This supports the notion that physiological factors, such as hormonal influences, may shape microbial composition differently across sexes. Additionally, [32] noted the influence of age and gender on bacterial communities in Tibetan sheep, underscoring the complex role of biological factors in structuring microbiota.
Conversely, a significant influence of age on the beta diversity of intestinal fungi was observed, suggesting variability in fungal composition across different age groups. This result aligns with studies in other species, such as primates [33], where age-related fungal beta diversity was attributed to dietary changes and seasonal food availability, and human studies by [34], which identified age as a key factor in the mycobiota's structure. These findings reinforce the importance of both intrinsic biological factors and external environmental influences, such as diet and seasonal changes, in shaping the composition of microbial communities.”
(Lines 178–184)The percentages of unique and shared Amplicon Sequence Variants (ASVs) between sexes are given, but the biological implications of these percentages are not discussed. The overlap of 94.5% in bacteria, 96.4% in fungi, and 98.9% in archaea suggests minimal sex-specific differences, yet these findings aren't critically analyzed.
> Thank you for your comment, the information was added in results on line 205-209 “These results indicate a high degree of overlap in ASV composition between female and male cattle across all three microbial groups, with a small percentage of ASVs being sex-specific. Although the overlap is substantial, even small differences in ASV composition could still influence important physiological processes, such as metabolism or immune function.”
I was added in discussion in the line 535-543: “The high overlap in ASVs between female and male cattle across bacteria, fungi, and archaea suggests a shared core microbiome that likely supports essential microbial functions common to both sexes (Wu et al., 2024). Despite this similarity, the presence of a small subset of sex-specific ASVs, although limited in percentage, may contribute to distinct biological roles that align with sex-specific metabolic or immune processes (d’Afflitto et al., 2022). Such differences, while subtle, could influence physiological traits unique to each sex, underscoring the potential relevance of even minor variations in microbial composition (Shade et al., 2012). Further research may elucidate how these unique ASVs impact broader physiological or metabolic pathways.”
-d’Afflitto, M., Upadhyaya, A., Green, A., & Peiris, M. (2022). Association between sex hormone levels and gut microbiota composition and diversity—a systematic review. Journal of clinical gastroenterology, 56(5), 384-392.
-Shade, A., Peter, H., Allison, S. D., Baho, D. L., Berga, M., Bürgmann, H., ... & Handelsman, J. (2012). Fundamentals of microbial community resistance and resilience. Frontiers in microbiology, 3, 417.
-Wu, J., Shen, H., Lv, Y., He, J., Xie, X., Xu, Z., ... & Hou, X. (2024). Age over sex: evaluating gut microbiota differences in healthy Chinese populations. Frontiers in Microbiology, 15, 1412991.
(Lines 187–188)The p-values and F statistics presented are borderline significant (especially for sex in bacteria and archaea), but the power of the study (sample size) is not addressed. It’s possible that the results could be underpowered, which might explain the lack of significance in some cases.
> Thank you for your valuable observation. We apologize for not providing more detailed information on this point earlier. Below is a comprehensive explanation addressing the power and sample size considerations, as well as an analysis of the significance of our findings.
We used a preliminary sample of cattle to calculate the standard deviation for sex within each group, resulting in a sample standard deviation of 0.0952 for the index alpha-diversity observed variable. Based on these estimates, we established a significance level (α) of 0.05 and a study power (1-β) of 80%. The minimum detectable difference (MDD) in the observed variable was 0.55, consistent with values reported in relevant literature (You & Kim, 2021; Liang et al., 2022; Gaire et al., 2023; Bessegatto et al., 2024). Through sample size calculation using Infostat (Di Rienzo et al., 2010), we determined a sample size of 9:11 samples per sex group (a total of 20 samples), balancing statistical precision with operational feasibility.
In our genetic core, the selection process focuses on retaining animals best suited for breeding, which imposes constraints on the sample size. However, performing a power analysis and calculating the sample size based on preliminary data aligns with best practices in microbiome research, where variability can significantly impact results (La Rosa et al., 2012; Knight et al., 2018). As highlighted by Ferdous et al. (2022), power analyses in microbiome studies remain challenging due to the difficulty in estimating true effect sizes and microbiome composition. In response to your concern, we have added the following clarification in line 88: “A preliminary study was carried out to identify the ideal number of replicates needed for the research.”
In line with the findings from Ferdous et al. (2022), we conducted several analyses that demonstrate significant differences in microbial composition. Specifically, we observed significant differences in archaeal alpha diversity for both age and sex, as well as beta diversity for bacteria and archaea based on sex, and for fungi based on age. Although the p-values for bacteria (0.04) and archaea (0.03) are close to the 0.05 threshold, they remain statistically significant (Kers et al., 2022). This is further supported by the clear separation of groups in the PCoA plots. Moreover, we conducted a differential abundance analysis using EdgeR, which identified taxa associated with males and females, indicating taxonomic differences between the sexes.
While the statistical significance may be limited by the sample size, the results of the differential analysis and the observed separation in PCoA provide biological relevance to the findings. We appreciate your observation and hope this detailed explanation clarifies the strength of the results presented.
-Liang, Z., Zhang, J., Du, M., Ahmad, A. A., Wang, S., Zheng, J., ... & Ding, X. (2022). Age-dependent changes of hindgut microbiota succession and metabolic function of Mongolian cattle in the semi-arid rangelands. Frontiers in Microbiology, 13, 957341.
-Gaire, T. N., Scott, H. M., Noyes, N. R., Ericsson, A. C., Tokach, M. D., Menegat, M. B., ... & Volkova, V. V. (2023). Age influences the temporal dynamics of microbiome and antimicrobial resistance genes among fecal bacteria in a cohort of production pigs. Animal Microbiome, 5(1), 2.
-Bessegatto, J. A., Lisbôa, J. A. N., Santos, B. P., Curti, J. M., Montemor, C., Alfieri, A. A., ... & Costa, M. C. (2024). Fecal Microbial Communities of Nellore and Crossbred Beef Calves Raised at Pasture. Animals, 14(10), 1447.
-You, I., & Kim, M. J. (2021). Comparison of gut microbiota of 96 healthy dogs by individual traits: Breed, age, and body condition score. Animals, 11(8), 2432.
-Ferdous, T., Jiang, L., Dinu, I., Groizeleau, J., Kozyrskyj, A. L., Greenwood, C. M., & Arrieta, M. C. (2022). The rise to power of the microbiome: power and sample size calculation for microbiome studies. Mucosal Immunology, 15(6), 1060-1070.
-La Rosa, P. S., Brooks, J. P., Deych, E., Boone, E. L., Edwards, D. J., Wang, Q., ... & Shannon, W. D. (2012). Hypothesis testing and power calculations for taxonomic-based human microbiome data. PloS one, 7(12), e52078.
-Knight, R., Vrbanac, A., Taylor, B. C., Aksenov, A., Callewaert, C., Debelius, J., ... & Dorrestein, P. C. (2018). Best practices for analysing microbiomes. Nature Reviews Microbiology, 16(7), 410-422.
-Di Rienzo, J., Balzarini, M., Gonzalez, L., Casanoves, F., Tablada, M., & Walter Robledo, C. (2010). Infostat: software para análisis estadístico.
(Lines 213–221) The description of genera is purely descriptive without an interpretation of what the differences might mean in terms of functionality or cattle physiology. The presentation is also slightly redundant with the earlier section on phyla.
> Thank you for your comment. We acknowledge the need to avoid redundancy between the results and discussion sections. The functional interpretations of the differences in genera abundance have been thoroughly addressed in the discussion. In the results section, we will focus on presenting the data more succinctly to avoid overlap.
In line 400-413 “Consistent with previous research, the dominant bacterial phyla identified in the cattle gut microbiota were Firmicutes and Bacteroidota [35–39]. The fungal community was predominantly composed of the phylum Ascomycota, which aligns with findings from other studies in cattle [40,41]. These results suggest a stable fungal community structure at the phylum level, relatively unaffected by variations in environmental or dietary conditions [42,43].
In terms of archaeal composition, the identified phyla and genera were consistent with those reported in other ruminant studies. Similar to previous findings in cattle [30], the phylum Euryarchaeota and Halobacterota were predominant. This taxonomic pattern was also observed in other ruminants, such as moose, where [44] documented a similar predominance of these phyla in the rumen archaeal community. The consistency in the taxonomic composition of archaea and fungi across different ruminant species underscores the stability of these microbial groups in the intestinal microbiome, emphasizing the significant role of diet and environmental factors in shaping these communities.”
In the line 470-486 “Methanosphaera is a genus of methanogens that produces methane through methanol reduction, a process that requires less hydrogen compared to other methanogenic pathways [61]. This genus has been identified as one of the main contributors to methanogenesis in ruminants, especially in the solid fraction of the rumen, suggesting its relevance in methane production in livestock [62]. Recent studies indicate that the abundance of Methanosphaera tends to decrease with the inclusion of additives such as chitosan, which could represent a potential strategy for mitigating methane emissions; however, the effects observed so far in methane reduction have been limited [63]. In the study conducted, a significant negative correlation was found between Methanosphaera and the values of MCV, MCH, MON, and total protein in males. This suggests that Methanosphaera might be inversely related to these hematological and protein parameters, potentially indicating an adverse impact on host health and metabolism. Furthermore, since the abundance of Methanosphaera may be influenced by host genetics, it is possible that the mechanisms regulating hydrogen concentration and ruminal microbiota composition also affect these physiological parameters [64]. However, to fully understand how Methanosphaera and other methanogens affect the hematological and protein profiles in ruminants, further studies are needed to explore these complex interactions in more depth.”
(Lines 223–289) The text lacks a critical assessment of whether these correlations are robust or just statistically significant by chance.
> Thank you for your insightful comment. The issue of false positives was addressed by applying the False Discovery Rate (FDR) correction during the correlation analysis. This was implemented using the microeco package, which allows for the FDR adjustment. Specifically, correlations were calculated using the Spearman method, and the FDR correction was applied to control for multiple comparisons, ensuring the robustness of the results.
It was added in the line 146-149 “Spearman's rank correlation analyses, adjusted using False Discovery Rate (FDR) correction, were conducted to examine the associations between microbial genera and hematological parameters.”
-Liu, C., Cui, Y., Li, X., & Yao, M. (2021). microeco: an R package for data mining in microbial community ecology. FEMS microbiology ecology, 97(2), fiaa255.
Evaluate the study’s power to detect significant differences, especially where borderline significance is observed.
> Thank you for your insightful question regarding the power of the study to detect significant differences, particularly in cases of borderline significance.
To evaluate the power of the study, we performed a power analysis based on preliminary data. We calculated the sample size needed to achieve a power of 80% (1-β) with a significance level (α) of 0.05. Based on these calculations and using a sample standard deviation of 0.0952 for the observed variable, we determined a minimum detectable difference (MDD) of 0.55, which aligns with previous studies in the field (You & Kim, 2021; Liang et al., 2022; Gaire et al., 2023; Bessegatto et al., 2024). With this power, we aimed to detect biologically relevant differences between the groups while maintaining statistical precision.
However, in microbiome studies, calculating power is particularly challenging due to the high variability in microbiome composition and the difficulty in estimating true effect sizes (Ferdous et al., 2022). This is especially true when considering measures of beta diversity and taxonomic abundance, where the complexity of the data requires sophisticated methods that go beyond traditional power calculations (Kers et al., 2022). While the p-values for bacteria (0.04) and archaea (0.03) are near the threshold for significance, the results are still statistically significant and supported by the clear separation of groups in the PCoA plots.
Additionally, in response to the concern about borderline significance, it is important to note that we conducted a differential abundance analysis using EdgeR, which identified specific taxa associated with males and females, corroborating the findings from our beta diversity analysis. These results, combined with the visual separation in the PCoA plots, suggest that the detected differences are biologically meaningful despite the limited sample size.
In line 88 was added “A pilot study was conducted to determine the optimal replication number for the research.”
In conclusion, although the study’s power is appropriate based on the preliminary analysis and pilot data, we acknowledge that increasing the sample size in future studies could further enhance the power to detect more subtle differences, potentially lowering the p-values observed for bacteria and archaea.
-Liang, Z., Zhang, J., Du, M., Ahmad, A. A., Wang, S., Zheng, J., ... & Ding, X. (2022). Age-dependent changes of hindgut microbiota succession and metabolic function of Mongolian cattle in the semi-arid rangelands. Frontiers in Microbiology, 13, 957341.
-Gaire, T. N., Scott, H. M., Noyes, N. R., Ericsson, A. C., Tokach, M. D., Menegat, M. B., ... & Volkova, V. V. (2023). Age influences the temporal dynamics of microbiome and antimicrobial resistance genes among fecal bacteria in a cohort of production pigs. Animal Microbiome, 5(1), 2.
-Bessegatto, J. A., Lisbôa, J. A. N., Santos, B. P., Curti, J. M., Montemor, C., Alfieri, A. A., ... & Costa, M. C. (2024). Fecal Microbial Communities of Nellore and Crossbred Beef Calves Raised at Pasture. Animals, 14(10), 1447.
-You, I., & Kim, M. J. (2021). Comparison of gut microbiota of 96 healthy dogs by individual traits: Breed, age, and body condition score. Animals, 11(8), 2432.
-Ferdous, T., Jiang, L., Dinu, I., Groizeleau, J., Kozyrskyj, A. L., Greenwood, C. M., & Arrieta, M. C. (2022). The rise to power of the microbiome: power and sample size calculation for microbiome studies. Mucosal Immunology, 15(6), 1060-1070.
-La Rosa, P. S., Brooks, J. P., Deych, E., Boone, E. L., Edwards, D. J., Wang, Q., ... & Shannon, W. D. (2012). Hypothesis testing and power calculations for taxonomic-based human microbiome data. PloS one, 7(12), e52078.
-Knight, R., Vrbanac, A., Taylor, B. C., Aksenov, A., Callewaert, C., Debelius, J., ... & Dorrestein, P. C. (2018). Best practices for analysing microbiomes. Nature Reviews Microbiology, 16(7), 410-422.
-Di Rienzo, J., Balzarini, M., Gonzalez, L., Casanoves, F., Tablada, M., & Walter Robledo, C. (2010). Infostat: software para análisis estadístico.
-Kers, J. G., & Saccenti, E. (2022). The power of microbiome studies: some considerations on which alpha and beta metrics to use and how to report results. Frontiers in microbiology, 12, 796025.
Conduct multivariate analyses (e.g., Canonical Correspondence Analysis) to better understand how environmental or physiological factors interact with microbial diversity and composition.
> Thank you for your valuable suggestion. A Canonical Correspondence Analysis (CCA) has been added to better elucidate the interactions between environmental and physiological factors with microbial diversity and composition. This multivariate approach provides deeper insights into how these variables influence the microbial community structure in our study.
Figure S3: Canonical correlation analysis for Sex. A) Bacteria CCA. B) Fungal CCA. C) Archaea CCA
It was added in results in the line 407-417 “Canonical Correspondence Analysis (CCA) (Fig. S3) was conducted to explore the associations between microbial composition and hematological parameters, differentiated by sex. With respect to bacteria (Fig. S3A), females appeared to be more associated with parameters such as WBC, HGB, MCV, and MCH, whereas males showed a potential relationship with Succinivibrionaceae_UCG-002 and Clostridium sensu stricto 3. Regarding fungi (Fig. S3B), females seemed to be linked to Zaanenomyces, which was associated with parameters like HGB and TP, while males displayed a possible association with Cadophora and variables such as PLT and NEU%. In the case of archaea (Fig. S3C), males demonstrated a notable association with methanogenic taxa, including Methanobrevibacter and Methanosphaera, while females exhibited correlations with MCH, MCV, and MON.”
It was added in discussion in the line 760-772 “In the CCA analysis, the differences observed between healthy males and females in the parameters EOS, EOS% and BAS% suggest that the intestinal microbiota could play a key role in the regulation of inflammation and immune response. In males, higher levels of eosinophils and basophils could be linked to immune modulation, possibly through the interaction between the microbiota and the immune system, which regulates inflammatory mediators and maintains immune homeostasis (Ondari et al., 2021; Wu et al., 2024). In contrast, females showed a stronger relationship with fermentative bacteria such as Succinivibrionaceae_UCG-002 and Clostridium sensu stricto 3, known for their role in the production of short chain fatty acids (SCFA), suggesting an adaptation of their microbiota towards the energy metabolism and fermentation (Holman et al., 2019; Cheng et al., 2022). These differences indicate a possible metabolic and immunological specialization influenced by sex and the intestinal microbiota in cattle.”
-Holman, D. B., & Gzyl, K. E. (2019). A meta-analysis of the bovine gastrointestinal tract microbiota. FEMS microbiology ecology, 95(6), fiz072.
-Cheng, J.; Zhang, X.; Xu, D.; Zhang, D.; Zhang, Y.; Song, Q.; Li, X.; Zhao, Y.; Zhao, L.; Li, W.; et al. Relationship between Rumen Microbial Differences and Traits among Hu Sheep, Tan Sheep, and Dorper Sheep. J. Anim. Sci. 2022, 100, skac261.
-Ondari, E., Calvino-Sanles, E., First, N. J., & Gestal, M. C. (2021). Eosinophils and bacteria, the beginning of a story. International Journal of Molecular Sciences, 22(15), 8004.
-Wu, D., Zhao, P., Wang, C., Huasai, S., Chen, H., & Chen, A. (2024). Differences in the intestinal microbiota and association of host metabolism with hair coat status in cattle. Frontiers in Microbiology, 15, 1296602.
Include functional predictions (e.g., through PICRUSt2) to see how the observed microbial shifts might translate to functional changes in metabolism or immune function.
> Thank you for the insightful suggestion. Functional predictions were indeed included through PICRUSt2 to explore how the observed microbial shifts might translate into functional changes, particularly regarding metabolic and immune-related pathways. This approach provides a clearer understanding of the potential metabolic and immunological impacts of microbial community changes within our study.
Figure S4: Prediction and comparison of bacterial community functions between sexes. PICRUSt2 was employed to infer changes in the composition of bacterial functional groups. STAMP software was utilized to assess the differences in KEGG functions between females and males, using Welch's two-sided t-test.
It was added in results in the line 418-426 “The analysis performed with PICRUSt (Fig. S4) revealed significant differences in several predicted metabolic functions between males and females. In males, higher proportions were identified in key metabolic pathways, such as purine nucleotide de novo biosynthesis (DENOVOPURINE2-PWY) and D-galacturonate degradation (GALACTUROCAT-PWY), suggesting increased microbial activity in these routes associated with nucleotide and carbohydrate metabolism. On the other hand, females showed higher proportions in pathways related to starch degradation (PWY-6731), geranylgeranyl diphosphate biosynthesis (PWY-5910), and photorespiration (PWY-181).”
It was added in discussion in the line 773-788 “The increased activity of the GALACTUROCAT-PWY pathway in male cattle suggests a microbiological adaptation that enhances the breakdown of complex carbohydrates, such as pectin, optimizing energy metabolism through the production of short-chain fatty acids (SCFAs), which are essential for intestinal and energy homeostasis (Yao et al., 2023; Zhang et al., 2023). This intensified activity in males may reflect a physiological adjustment of the microbiota to meet sex-specific energy demands, promoting gut health and nutrient use efficiency (Wang et al., 2024). The increased activity of the PWY-5265 pathway (peptidoglycan biosynthesis) in female cattle suggests an adaptation that enhances bacterial cell wall stability, supporting immune regulation and mucosal integrity, and promoting a more resilient microbiota (Yu et al., 2023; Wang et al., 2023). The presence of Faecalibacterium in females aligns with this function, given its role in butyrate production and immune modulation, which contribute to intestinal health. Previous studies associate Faecalibacterium prevalence with improved weight gain and reduced diarrhea in early-life stages, underscoring its probiotic potential (69–71). The detection of Faecalibacterium here highlights its positive impact on metabolic balance and welfare in female cattle.”
-Yao, H., Williams, B. A., Mikkelsen, D., Flanagan, B. M., & Gidley, M. J. (2023). Composition and functional profiles of human faecal microbiota fermenting plant-based food particles are related to water-holding capacity more than particle size. Food Hydrocolloids, 141, 108714.
-Zhang, L., Tepes, M., Tong, A., & Lee, D. (2023). Microbial diversity of Smokers is not influenced by dietary Fiber intake although smoking alters functional Pathway abundances. Undergraduate Journal of Experimental Microbiology and Immunology, 28.
-Wang, D., Wang, X., Han, J., You, C., Liu, Z., & Wu, Z. (2024). Effect of Lacticaseibacillus casei LC2W Supplementation on Glucose Metabolism and Gut Microbiota in Subjects at High Risk of Metabolic Syndrome: A Randomized, Double-blinded, Placebo-controlled Clinical Trial. Probiotics and Antimicrobial Proteins, 1-15.
-Wang, M. W., Ma, W. J., Wang, Y., Ma, X. H., Xue, Y. F., Guan, J., & Chen, X. (2023). Comparison of the effects of probiotics, rifaximin, and lactulose in the treatment of minimal hepatic encephalopathy and gut microbiota. Frontiers in Microbiology, 14, 1091167.
-Yu, S., Ge, X., Xu, H., Tan, B., Tian, B., Shi, Y., ... & Qian, J. (2023). Gut microbiome and mycobiome in inflammatory bowel disease patients with Clostridioides difficile infection. Frontiers in Cellular and Infection Microbiology, 13, 1129043.

Reviewer 2 Report
Comments and Suggestions for Authors
This study has investigated sex-induced changes in the microbiota of Simmental cattle. It also emphasizes the importance of gender selection in livestock studies involved with gut microbiota. Overall, this is an interesting and important study. Following are some comments on the manuscript.
Title: It is suggested to be changed into “…. in Gastrointestinal Tract of Simmental cattle”.
Line 84: “Previously DNA samples were obtained from Richard Estrada et al. (2024).” This sentence is confusing. No reference was provided. Were the samples obtained from a previous study?
Line 133-135: What was the statistical analysis method of alpha diversity? T test?
Figure 1: It is not necessary to indicate a and b in the Figure with only 2 groups. What is Y and S? P value is suggested to be shown solely in the Figure.
Line 164-169: Figure 2 is about the beta diversity based on sex. However, in the text, the authors showed significant difference on year indicating with Fig.2B and Fig.2C. These should be changed into Table 1.
Figure 5: The abbreviations of hematological parameters should be explained in the Figure caption and in the relevant text when it first appears. Same with Figure 7 and Table 2.
Discussion: The discussion is too long. Please reduce the length of discussion. Many results should not be repeated in the discussion part.
Line 367-369: In this study, both alpha diversity and beta diversity (Fig 1 and 2) have been shown to be changed significantly in the gut microbiota between different sexes. Table 1 clearly showed a significant difference on the beta diversity of bacteria between females and males. Why the authors indicate “no significant differences related to age or gender”?
Line 383-385: Herein, the authors have indicated “significant gender-based differences in the beta diversity of bacteria and archaea”. The contradiction between paragraphs is really confusing.
Discussion: The authors have discussed a lot about certain genus species. Is there any relationship between there genera? What implications could be achieved on the cattle farming industry based on the present results?
Discussion: Sex-induced changes on the hematological parameters might lead to alterations on the gut microbiota between females and males. Is there any difference on the hematological parameters of Simmental cattle between females and males?
Comments on the Quality of English Language
Moderate editing of English language required.
Author Response
This study has investigated sex-induced changes in the microbiota of Simmental cattle. It also emphasizes the importance of gender selection in livestock studies involved with gut microbiota. Overall, this is an interesting and important study. Following are some comments on the manuscript.
> Thank you for your positive feedback and for acknowledging the significance of our study. We appreciate your valuable comments and suggestions, which will help to further improve the manuscript.
Title: It is suggested to be changed into “…. in Gastrointestinal Tract of Simmental cattle”.
> Thank you for your suggestion. The title change to “... in Gastrointestinal Tract of Simmental cattle” has been accepted and implemented.
Line 84: “Previously DNA samples were obtained from Richard Estrada et al. (2024).” This sentence is confusing. No reference was provided. Were the samples obtained from a previous study?
> Thank you for your comment. We apologize for the oversight. The sentence refers to DNA samples that were previously obtained and published as part of the study by Richard Estrada et al. (2024). We will revise the text to include the correct citation and ensure clarity in the manuscript.
In the line 84 “Previously, DNA samples were obtained in a prior study (Richard Estrada et al., 2024).”
Line 133-135: What was the statistical analysis method of alpha diversity? T test?
> Thank you for your comment. The statistical method for analyzing alpha diversity was indeed a two-way ANOVA, as mentioned in the text. For clarity, we have revised the phrase to ensure better understanding.
It was added in methods in the line 140-143 “Alpha diversity metrics for intestinal bacteria, such as observed species count (Observed), species richness estimate (ACE), Fisher’s index, and phylogenetic diversity (PD), were calculated, and a two-way ANOVA was applied to analyze the impact of age and sex”
Figure 1: It is not necessary to indicate a and b in the Figure with only 2 groups. What is Y and S? P value is suggested to be shown solely in the Figure.
> Thank you for your valuable feedback. We will remove the “a” and “b” labels from the figure, as suggested. The variables “Y” and “S” refer to Year and Sex, respectively, and we will now display only the p-value in the figure. Additionally, we have updated the figure description to clarify these variables. The updated caption will now read:
“Figure 1. Comparison of alpha diversity of archaea in the cattle gut microbiome between females and males. Y = Year, S = Sex. A) Observed. B) ACE. C) Fisher. D) PD.”
Line 164-169: Figure 2 is about the beta diversity based on sex. However, in the text, the authors showed significant difference on year indicating with Fig.2B and Fig.2C. These should be changed into Table 1.
> Thank you for your observation. We have revised Figure 2 to now display both age and sex, as indicated.
Figure 5: The abbreviations of hematological parameters should be explained in the Figure caption and in the relevant text when it first appears. Same with Figure 7 and Table 2.
> Thank you for your suggestion. A supplementary table 2 has been added providing explanations for all abbreviations of the hematological parameters in Figures 5, 7, and Table 2.
Discussion: The discussion is too long. Please reduce the length of discussion. Many results should not be repeated in the discussion part.
> Thank you for your observation. Discussion has been improved. In the line 439-468 was before “In this study, the analysis of bacterial and fungal alpha diversity in the cattle gut microbiota revealed no significant differences related to age or gender, aligning with findings from other animal species. For example, recent studies in musk deer and Tibetan goats, as well as ruminants from the Tibetan Plateau, have similarly reported no significant differences in alpha diversity indices such as Chao1 and Shannon when comparing individuals of different genders and age groups [26–28]. This suggests that age and gender may not substantially impact alpha diversity microbial diversity, with factors such as diet and environment likely exerting a greater influence.
However, the current study did identify significant differences in archaea alpha diversity within the intestinal microbiome of cattle based on gender, consistent with observations in other species. For instance, [29] reported notable differences in methanogenic archaea between young and adult reindeer in the Arctic tundra. Similarly, [30] found that the composition and diversity of microbial communities in the rumen during early life stages are significantly influenced by diet and environment. These findings underscore the importance of considering age, gender, and environmental factors as crucial determinants in the composition of the archaeal community.”
It was improved by “In this study, the analysis of bacterial and fungal alpha diversity in the cattle gut microbiota revealed no significant differences related to age or gender, aligning with findings from other animal species. For example, recent studies in musk deer and Tibetan goats, as well as ruminants from the Tibetan Plateau, have similarly reported no significant differences in alpha diversity indices such as Chao1 and Shannon when comparing individuals of different genders and age groups [26–28]. The indices used in this study—Observed, ACE, Fisher, and PD—were specifically chosen to capture different aspects of microbial diversity: richness (Observed and ACE) and phylogenetic diversity (PD), with Fisher providing additional insight into community diversity structure. These indices were selected to provide a robust evaluation of microbial diversity across groups.
This suggests that age and gender may not substantially impact microbial diversity in bacteria and fungi, with factors such as diet and environment likely exerting a greater influence. The lack of significant findings for bacteria and fungi may reflect a more generalized response to environmental factors in these communities, contrasting with archaea, which could be more sensitive to intrinsic physiological factors, such as hormonal differences between sexes.
However, the current study did identify significant differences in archaea alpha diversity within the intestinal microbiome of cattle based on gender, consistent with observations in other species. This variation in archaeal diversity by sex could be attributed to physiological factors that uniquely influence archaeal communities, potentially through mechanisms associated with hormone regulation or energy metabolism. For instance, [29] reported notable differences in methanogenic archaea between young and adult reindeer in the Arctic tundra. Similarly, [30] found that the composition and diversity of microbial communities in the rumen during early life stages are significantly influenced by diet and environment. These findings underscore the importance of considering a multifactorial approach, where age, gender, and environmental factors act as key determinants in shaping the archaeal community composition in the gut microbiome.”
In the line 439-468 was before “Moreover, the study identified significant gender-based differences in the beta diversity of bacteria and archaea, indicating variations in microbial community composition between males and females. This is consistent with previous research that has demonstrated gender-related differences in the structure of the intestinal microbiota across various species. For instance, [31] observed distinct clustering of bacterial communities in bamboo rats based on gender, while [26] reported differences in the beta diversity of bacteria and archaea in cattle according to gender, but not age. [32] also highlighted the influence of age and gender on the taxonomic composition of bacterial communities in the rumen of Tibetan sheep, further emphasizing the significance of these factors in microbiota structuring.
Additionally, a significant influence of age on the beta diversity of the intestinal mycobiota was observed, indicating variations in fungal composition across different age groups. This finding aligns with studies in other species, such as those by [33] in primates, which reported significant separation in fungal beta diversity related to age. This variability was attributed to dietary differences and seasonal food availability, a pattern similarly observed in human studies by [34], where age was identified as a determining factor in the structure of the intestinal mycobiota.”
It was improved by “Moreover, the study identified significant sex-based differences in the beta diversity of bacteria and archaea, indicating distinct microbial community compositions between males and females. This finding is consistent with previous research that has demonstrated gender-related differences in intestinal microbiota across various species. For instance, [31] observed distinct clustering of bacterial communities in bamboo rats based on gender, while [26] found similar patterns in cattle, reporting gender-specific differences in the beta diversity of bacteria and archaea. This supports the notion that physiological factors, such as hormonal influences, may shape microbial composition differently across sexes. Additionally, [32] noted the influence of age and gender on bacterial communities in Tibetan sheep, underscoring the complex role of biological factors in structuring microbiota.
Conversely, a significant influence of age on the beta diversity of intestinal fungi was observed, suggesting variability in fungal composition across different age groups. This result aligns with studies in other species, such as primates [33], where age-related fungal beta diversity was attributed to dietary changes and seasonal food availability, and human studies by [34], which identified age as a key factor in the mycobiota's structure. These findings reinforce the importance of both intrinsic biological factors and external environmental influences, such as diet and seasonal changes, in shaping the composition of microbial communities.”
In the line 439-468 was before “Previous studies have demonstrated that the administration of Saccharomyces boulardii can significantly alter the composition of the intestinal microbiota, reducing systemic and hepatic inflammation in obese and type 2 diabetic mouse models, as well as alleviating the symptoms of DSS-induced colitis, thereby improving the function of the intestinal barrier [52–55]. In other species, such as horses and cattle, Saccharomyces cerevisiae has been observed to influence the intestinal microbiota, particularly in the modulation of fiber-fermenting and lactate-stabilizing bacteria, suggesting a possible effect on digestion and intestinal health [56,57]. In the context of my study with bulls, a significant positive correlation of Saccharomyces with RBC, WBC, and LYM% counts was identified, as well as a negative correlation with protein levels. These results indicate that Saccharomyces could be modulating the intestinal microbiota in a way that impacts hematological parameters, which could influence the health and management of cattle through mechanisms that optimize metabolic balance and digestive function [58]. The ability of Saccharomyces to influence these variables may have important implications for improving production outcomes and health in livestock production.
Aspergillus is a genus that frequently inhabits the gastrointestinal tract of various animals, where it contributes to beneficial processes, such as fiber digestion [59]. It has been proposed that metabolites generated by Aspergillus help preserve a favorable microbial balance, especially when disease-triggering factors are not present [60]. The ability of this genus to positively interact with other microbial species, promoting a balanced and healthy gut microbiome [60], may also explain the significant positive correlation observed with platelet counts in cows. The presence of Aspergillus in a healthy microbiome could indirectly influence the regulation of hematological parameters, such as platelets, highlighting its broader role in host physiological homeostasis.
Methanosphaera is a genus of methanogens that produces methane through methanol reduction, a process that requires less hydrogen compared to other methanogenic pathways [61]. This genus has been identified as one of the main contributors to methanogenesis in ruminants, especially in the solid fraction of the rumen, suggesting its relevance in methane production in livestock [62]. Recent studies indicate that the abundance of Methanosphaera tends to decrease with the inclusion of additives such as chitosan, which could represent a potential strategy for mitigating methane emissions; however, the effects observed so far in methane reduction have been limited [63]. In the study conducted, a significant negative correlation was found between Methanosphaera and the values of MCV, MCH, MON, and total protein in males. This suggests that Methanosphaera might be inversely related to these hematological and protein parameters, potentially indicating an adverse impact on host health and metabolism. Furthermore, since the abundance of Methanosphaera may be influenced by host genetics, it is possible that the mechanisms regulating hydrogen concentration and ruminal microbiota composition also affect these physiological parameters [64]. However, to fully understand how Methanosphaera and other methanogens affect the hematological and protein profiles in ruminants, further studies are needed to explore these complex interactions in more depth.
The identification of Prevotellaceae_UCG-004 in male bulls is particularly noteworthy due to its prominent role in ruminal fermentation and the production of volatile fatty acids (VFAs) such as acetate and butyrate [65]. Previous research has demonstrated a significant association between Prevotellaceae_UCG-004 and enhanced growth performance in high body weight goat kids, attributed to its ability to degrade complex polysaccharides and increase VFA production, thereby promoting ruminal epithelium development [65]. Additionally, Prevotellaceae_UCG-004 has been linked to ascorbate and aldarate metabolism in sheep, suggesting that a higher dietary vitamin C content could enhance its proportion in the ruminal microbiota [66]. The presence of Prevotellaceae_UCG-004 in male bulls may similarly contribute to optimizing energy metabolism, indicating a potential role for this bacterium in improving productive performance and health. These findings underscore the need for further research to explore how Prevotellaceae_UCG-004 could be utilized to develop targeted nutritional strategies that maximize growth and efficiency in livestock production.
The co-occurrence and functional roles of genera such as Methanobrevibacter, Prevotellaceae_UCG-004, and Treponema highlight a complex and interdependent microbial ecosystem within the bovine gut. Methanobrevibacter species, prominent across all age groups, contribute to methane production by metabolizing hydrogen generated during ruminal fermentation, a process essential for maintaining a stable anaerobic environment conducive to fermentation [61, 62]. The mutualistic interactions between Methanobrevibacter and fiber-degrading bacteria, including Prevotellaceae_UCG-004, support energy optimization through volatile fatty acid (VFA) production, which enhances cattle growth and performance [65, 66]. Treponema, associated with favorable hematological profiles, may further complement this ecosystem by modulating nutrient metabolism and maintaining microbial stability [45, 46]. The integrated roles of these microbial genera suggest potential pathways for targeted interventions to improve ruminant health and productivity. By harnessing and modulating these microbial networks through dietary adjustments or microbial supplements, the cattle farming industry could achieve advancements in methane mitigation, optimized growth, and overall animal welfare.
The study of Negativibacillus has demonstrated its involvement in intestinal dysbiosis, particularly in calves infected with EHEC O157, suggesting its role in altering intestinal homeostasis and overall host health [67]. The identification of Negativibacillus in the microbiota of male bulls could indicate its relevance to intestinal health and productive performance. Although its presence has been associated with dysbiosis in other contexts, in healthy bulls, it might be linked to a robust microbiome that supports overall well-being. Additionally, the importance of Negativibacillus in reproductive health has been highlighted, suggesting that this genus could significantly impact health and reproductive performance in livestock [68].
The presence of the genus Faecalibacterium in female cattle is significant due to its well-documented role in butyrate production and immune system modulation [69]. Furthermore, the prevalence of Faecalibacterium during the first week of life in calves has been associated with greater weight gain and a lower incidence of diarrhea, suggesting its potential as a beneficial probiotic in animal production [70]. Recent studies also indicate that the underrepresentation of Faecalibacterium may be related to imbalances in gut microbiota, potentially affecting animal health and performance [71]. The detection of Faecalibacterium in this study suggests a positive impact on the intestinal health and metabolic balance of the females, highlighting its potential contribution to their welfare and performance.
The detection of the genus Naganishia in bulls is of particular interest due to its potential role in rumen health and interaction with other microorganisms within the rumen ecosystem. Previous studies have identified Naganishia in various environments, including the rumen of young ruminants, where it may play a role in fiber digestion and modulation of the rumen microbial environment [72]. Moreover, Naganishia has been observed to be more abundant in healthy piglets compared to those with diarrhea, suggesting a potential protective role in maintaining intestinal microbial balance and preventing gastrointestinal pathologies [73].”
It was improved by “Studies indicate that Saccharomyces boulardii modulates the intestinal microbiota, reducing systemic and hepatic inflammation in models of obesity and type 2 diabetes, and enhancing the intestinal barrier function in cases of induced colitis [52-55]. In species like horses and cattle, Saccharomyces cerevisiae also impacts the microbiota, modulating bacteria associated with fiber fermentation and lactate stabilization [56, 57]. In this study, a positive correlation was identified between Saccharomyces and RBC, WBC, and LYM% counts, with a negative correlation with protein levels, suggesting that this genus may influence the intestinal microbiota and impact hematological parameters essential to cattle health [58].
Aspergillus, a common genus in the gastrointestinal tract of various animals, is linked to fiber digestion and microbial balance preservation, especially in the absence of pathogenic factors. Here, a significant positive correlation emerged between Aspergillus and platelet counts, suggesting an indirect role in regulating hematological parameters [59, 60]. Methanosphaera, recognized for its role in methanogenesis through methanol reduction, contributes to methane production in ruminants, primarily in the solid fraction of the rumen [61, 62]. In this analysis, a significant negative correlation was observed between Methanosphaera and values of MCV, MCH, MON, and protein levels in males, potentially indicating an adverse impact on host health and metabolism. This highlights the importance of investigating complex interactions between methanogens and physiological parameters [63, 64].
The identification of Prevotellaceae_UCG-004 in bulls underscores its role in ruminal fermentation and volatile fatty acid (VFA) production, particularly acetate and butyrate. Prior studies associate Prevotellaceae_UCG-004 with improved growth performance in high-body-weight goat kids, attributed to its ability to degrade complex polysaccharides and support ruminal epithelium development [65]. Additionally, its association with ascorbate and aldarate metabolism suggests that a high dietary vitamin C content could enhance its presence in the ruminal microbiota, optimizing energy metabolism and productive performance in cattle [66]. The co-occurrence of genera such as Methanobrevibacter, Prevotellaceae_UCG-004, and Treponema suggests an interdependent microbial network that optimizes the rumen’s anaerobic environment, promoting VFA production essential for cattle growth and performance. These mutualistic interactions, especially between Methanobrevibacter and fiber-degrading bacteria, highlight opportunities for dietary interventions to improve rumen health and potentially reduce methane emissions [61, 65, 66].
The presence of Negativibacillus in the bull microbiota could indicate a robust microbiome that supports overall well-being. Although associated with dysbiosis in other contexts, in healthy animals, it may contribute to intestinal homeostasis and productive performance, with additional positive effects on reproductive health [67, 68].
The detection of Naganishia in bulls suggests a role in rumen health and modulation of the ruminal microbial environment. Prior studies have identified Naganishia in young ruminants, where it may support fiber digestion and microbial balance, contributing to rumen ecosystem stability [72, 73].”
In the line 439-468 was before “Methanobrevibacter ruminantium was identified in the female subjects, consistent with research linking this species to lower methane emissions in bovines [77]. Prior studies have shown that M. ruminantium is more prevalent in cattle with lower methane production, compared to Methanobrevibacter gottschalkii, which is associated with higher emissions [78]. The increased presence of M. ruminantium in females may indicate an adaptive mechanism within the microbiome that optimizes hydrogen and formate utilization, thereby reducing methane production. This observation suggests that the females in this study could exhibit reduced methane emissions, aligning with findings from other studies that correlate M. ruminantium abundance with decreased methane output in ruminants [79].”
It was improved by “Methanobrevibacter ruminantium, identified in females, is associated with lower methane emissions compared to Methanobrevibacter gottschalkii, potentially indicating an adaptation of the ruminal microbiota towards optimized hydrogen and formate utilization in methane metabolism [77-79]”
Finally was necessary added a paragraph for the CCA and Picrust analysis
It was added in discussion in the line 760-772 “In the CCA analysis, the differences observed between healthy males and females in the parameters EOS, EOS% and BAS% suggest that the intestinal microbiota could play a key role in the regulation of inflammation and immune response. In males, higher levels of eosinophils and basophils could be linked to immune modulation, possibly through the interaction between the microbiota and the immune system, which regulates inflammatory mediators and maintains immune homeostasis (Ondari et al., 2021; Wu et al., 2024). In contrast, females showed a stronger relationship with fermentative bacteria such as Succinivibrionaceae_UCG-002 and Clostridium sensu stricto 3, known for their role in the production of short chain fatty acids (SCFA), suggesting an adaptation of their microbiota towards the energy metabolism and fermentation (Holman et al., 2019; Cheng et al., 2022). These differences indicate a possible metabolic and immunological specialization influenced by sex and the intestinal microbiota in cattle.”
-Holman, D. B., & Gzyl, K. E. (2019). A meta-analysis of the bovine gastrointestinal tract microbiota. FEMS microbiology ecology, 95(6), fiz072.
-Cheng, J.; Zhang, X.; Xu, D.; Zhang, D.; Zhang, Y.; Song, Q.; Li, X.; Zhao, Y.; Zhao, L.; Li, W.; et al. Relationship between Rumen Microbial Differences and Traits among Hu Sheep, Tan Sheep, and Dorper Sheep. J. Anim. Sci. 2022, 100, skac261.
-Ondari, E., Calvino-Sanles, E., First, N. J., & Gestal, M. C. (2021). Eosinophils and bacteria, the beginning of a story. International Journal of Molecular Sciences, 22(15), 8004.
-Wu, D., Zhao, P., Wang, C., Huasai, S., Chen, H., & Chen, A. (2024). Differences in the intestinal microbiota and association of host metabolism with hair coat status in cattle. Frontiers in Microbiology, 15, 1296602.
It was added in discussion in the line 773-788 “The increased activity of the GALACTUROCAT-PWY pathway in male cattle suggests a microbiological adaptation that enhances the breakdown of complex carbohydrates, such as pectin, optimizing energy metabolism through the production of short-chain fatty acids (SCFAs), which are essential for intestinal and energy homeostasis (Yao et al., 2023; Zhang et al., 2023). This intensified activity in males may reflect a physiological adjustment of the microbiota to meet sex-specific energy demands, promoting gut health and nutrient use efficiency (Wang et al., 2024). The increased activity of the PWY-5265 pathway (peptidoglycan biosynthesis) in female cattle suggests an adaptation that enhances bacterial cell wall stability, supporting immune regulation and mucosal integrity, and promoting a more resilient microbiota (Yu et al., 2023; Wang et al., 2023). The presence of Faecalibacterium in females aligns with this function, given its role in butyrate production and immune modulation, which contribute to intestinal health. Previous studies associate Faecalibacterium prevalence with improved weight gain and reduced diarrhea in early-life stages, underscoring its probiotic potential (69–71). The detection of Faecalibacterium here highlights its positive impact on metabolic balance and welfare in female cattle.”
-Yao, H., Williams, B. A., Mikkelsen, D., Flanagan, B. M., & Gidley, M. J. (2023). Composition and functional profiles of human faecal microbiota fermenting plant-based food particles are related to water-holding capacity more than particle size. Food Hydrocolloids, 141, 108714.
-Zhang, L., Tepes, M., Tong, A., & Lee, D. (2023). Microbial diversity of Smokers is not influenced by dietary Fiber intake although smoking alters functional Pathway abundances. Undergraduate Journal of Experimental Microbiology and Immunology, 28.
-Wang, D., Wang, X., Han, J., You, C., Liu, Z., & Wu, Z. (2024). Effect of Lacticaseibacillus casei LC2W Supplementation on Glucose Metabolism and Gut Microbiota in Subjects at High Risk of Metabolic Syndrome: A Randomized, Double-blinded, Placebo-controlled Clinical Trial. Probiotics and Antimicrobial Proteins, 1-15.
-Wang, M. W., Ma, W. J., Wang, Y., Ma, X. H., Xue, Y. F., Guan, J., & Chen, X. (2023). Comparison of the effects of probiotics, rifaximin, and lactulose in the treatment of minimal hepatic encephalopathy and gut microbiota. Frontiers in Microbiology, 14, 1091167.
-Yu, S., Ge, X., Xu, H., Tan, B., Tian, B., Shi, Y., ... & Qian, J. (2023). Gut microbiome and mycobiome in inflammatory bowel disease patients with Clostridioides difficile infection. Frontiers in Cellular and Infection Microbiology, 13, 1129043.
Line 367-369: In this study, both alpha diversity and beta diversity (Fig 1 and 2) have been shown to be changed significantly in the gut microbiota between different sexes. Table 1 clearly showed a significant difference on the beta diversity of bacteria between females and males. Why the authors indicate “no significant differences related to age or gender”?
> Thank you for your valuable comment. We have carefully reviewed the text and clarified the distinction between alpha and beta diversity in the manuscript. The paragraph now reflects that no significant differences were observed in alpha diversity for bacteria and fungi with respect to age or gender, while significant differences were found in beta diversity, particularly between males and females. Additionally, we have emphasized the findings related to archaeal alpha diversity, where gender-specific differences were noted. These revisions ensure consistency with the results presented in Figures 1, 2, and Table 1.
Line 383-385: Herein, the authors have indicated “significant gender-based differences in the beta diversity of bacteria and archaea”. The contradiction between paragraphs is really confusing.
> Thank you for your comment. We have reviewed the relevant sections to resolve any potential contradictions. The paragraph now clarifies that significant gender-based differences were observed in the beta diversity of bacteria and archaea, while no significant differences were found in alpha diversity for bacteria and fungi. The text has been revised to ensure consistency and to accurately reflect the observed results. We apologize for any confusion and have made the necessary corrections.
Discussion: The authors have discussed a lot about certain genus species. Is there any relationship between there genera? What implications could be achieved on the cattle farming industry based on the present results?
> Thanks for your suggestion. Added a discussion paragraph for correlations between microorganisms in line 592-601 "Additionally, its association with ascorbate and aldarate metabolism suggests that a high dietary vitamin C content could enhance its presence in the ruminal microbiota, optimizing energy metabolism and productive performance in cattle [66]. The co-occurrence of genera such as Methanobrevibacter, Prevotellaceae_UCG-004, and Treponema suggests an interdependent microbial network that optimizes the rumen’s anaerobic environment, promoting VFA production essential for cattle growth and performance. These mutualistic interactions, especially between Methanobrevibacter and fiber-degrading bacteria, highlight opportunities for dietary interventions to improve rumen health and potentially reduce methane emissions [61, 65, 66].", and improved the conclusion to highlight useful farming of microorganisms in the L803-813 "The study revealed significant gender-based differences in the gut microbiota of bovines, with distinct patterns in bacterial, archaeal, and fungal communities between males and females. The observed negative correlation between Methanosphaera and key hematological parameters in males suggests its influence on metabolic processes, which could impact health and performance. Additionally, the association of Treponema with favorable hematological profiles in males, and the correlation of Akkermansia with hematological variations in females, emphasize the relevance of gender in shaping the microbiota. These insights underscore the potential of tailoring microbiota management strategies based on gender to optimize cattle health and productivity, representing a promising approach for enhancing performance and efficiency in the cattle farming industry."
Discussion: Sex-induced changes on the hematological parameters might lead to alterations on the gut microbiota between females and males. Is there any difference on the hematological parameters of Simmental cattle between females and males?
> Thank you for your insightful comment. We have now conducted a Mann-Whitney statistical test to evaluate the differences in hematological parameters between females and males of Simmental cattle. The results are presented in Supplementary Table 8, and this has been included in the description within the Results section for further clarity.
In the line 351-354 “A Mann-Whitney test was conducted to assess the differences in hematological parameters between males and females. Significant differences based on sex were observed in leukocytes, neutrophils, segmented, lymphocytes, neutrophils (%), and lipids (Table S8).”
It was added in discussion in the line 543-548 “The significant differences in hematological parameters between sexes suggest variations in immune activity and metabolic profile, which could be related to specific physiological demands or differences in hormonal regulation between females and males (Ortona et al., 2019; Shepherd et al., 2021). These findings highlight the potential impact of sex on health and response to external conditions, justifying additional studies to better understand these influences.”
-Shepherd, R., Cheung, A. S., Pang, K., Saffery, R., & Novakovic, B. (2021). Sexual dimorphism in innate immunity: the role of sex hormones and epigenetics. Frontiers in immunology, 11, 604000.
-Ortona, E., Pierdominici, M., & Rider, V. (2019). Sex hormones and gender differences in immune responses. Frontiers in immunology, 10, 1076.
Round 2
Reviewer 1 Report
Comments and Suggestions for Authors
the authors have addressed the feedback and incorporated the suggested revisions into the manuscript and I believe it now meets the standards required for publication. I think that the revised manuscript is now ready for consideration for publication
Reviewer 2 Report
Comments and Suggestions for Authors
The manuscript has been improved.